# Implicit Neural Spatial Representations for Time-dependent PDEs

## Abstract

Numerically solving partial differential equations (PDEs) often entails spatial and temporal discretizations. Traditional methods (e.g., finite difference, finite element, smoothed-particle hydrodynamics) frequently adopt *explicit* spatial discretizations, such as grids, meshes, and point clouds, where each degree-of-freedom corresponds to a location in space. While these explicit spatial correspondences are intuitive to model and understand, these representations are not necessarily optimal for accuracy, memory-usage, or adaptivity. In this work, we explore implicit neural representation as an alternative spatial discretization, where spatial information is *implicitly* stored in the neural network weights. With implicit neural spatial representation, PDE-constrained time-stepping translates into updating neural network weights, which naturally integrates with commonly adopted optimization time integrators. Our approach requires neither training data nor training/testing separation. Our method is the solver itself, just like the classical PDE solver. We validate our approach on a variety of classic PDEs with examples involving large elastic deformations, turbulent fluids, and multi-scale phenomena. While slower to compute than traditional representations, our approach exhibits higher accuracy, lower memory consumption, and dynamically adaptive allocation of degrees of freedom without complex remeshing.

## 1 Introduction

Many science and engineering problems can be formulated as spatiotemporal partial differential equations (PDEs),

$$\mathcal{F}(\boldsymbol{f}, \boldsymbol{\nabla} \boldsymbol{f}, \boldsymbol{\nabla}^2 \boldsymbol{f}, \ldots, \dot{\boldsymbol{f}}, \ddot{\boldsymbol{f}}, \ldots) = \mathbf{0}, \quad \boldsymbol{f}(\boldsymbol{x}, t) : \Omega \times \mathcal{T} \to \mathbb{R}^d \,. \tag{1}$$

where $\Omega \in \mathbb{R}^m$ and $\mathcal{T} \in \mathbb{R}$ are the spatial and temporal domains, respectively. Examples include the inviscid Navier-Stokes equations for fluid dynamics and the elastodynamics equation for solid mechanics.

To numerically solve these PDEs, we oftentimes introduce temporal discretizations, $\{t_n\}_{n=0}^T$, where $T$ is the number of temporal discretization samples and $\Delta t = t_{n+1} - t_n$ is the time step size. The solution to Equation (1) then becomes a list of spatially dependent vector fields: $\{\boldsymbol{f}^n(\boldsymbol{x})\}_{n=0}^T$.

Traditional approaches represent these spatially dependent vector fields using grids, meshes, or point clouds. For example, the grid-based linear finite element method (Hughes, 2012) defines a shape function $N^i$ on each grid node and represents the spatially dependent vector field as $\boldsymbol{f}^n(\boldsymbol{x}) = \sum_{i=1}^P \boldsymbol{f}_i^n N^i$, where $P$ is the number of spatial samples.

While widely adopted in scientific computing applications, these traditional spatial representations are not without drawbacks:

1. Spatial discretization errors abound in fluid simulations as artificial numerical diffusion (Lantz, 1971), dissipation (Fedkiw et al., 2001), and viscosity (Roache, 1998). These errors also appear in solid simulations as inaccurate collision resolution (Müller et al., 2015) and numerical fractures (Sadeghirad et al., 2011).

2. Memory usage spikes with the number of spatial samples $P$ (Museth, 2013).

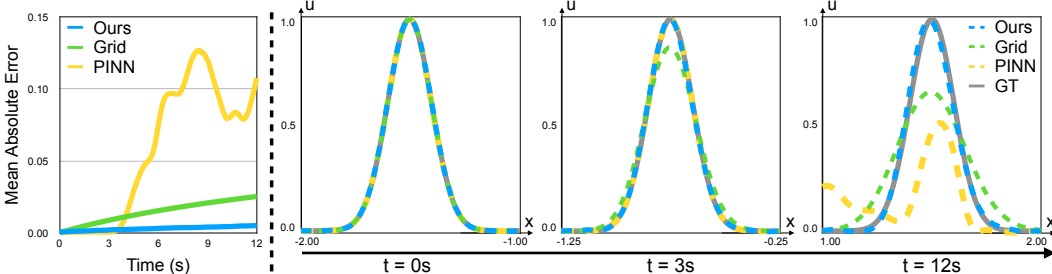

Figure 1: **1D advection example:** A Gaussian-shaped wave initially centered at $x = -1.5$ moves rightward with a constant velocity of $0.25$. From left to right, we show mean absolute error plot over time and solutions at $t = 0s$, $t = 3s$ and $t = 12s$, respectively. The solution from grid-based finite difference method (green) tends to diffuse over time. PINN (yellow), trained within temporal range $0 \sim 3s$, fails to generalize for $t = 12s$. Our solution (blue) approximates the ground truth (grey) the best over time. All three representations have the same memory footprint: our approach and PINN (Raissi et al., 2019) both use $\alpha = 2$ hidden layers of width $\beta = 20$, and the finite difference grid resolution is 901.

3. Adaptive meshing (Narain et al., 2012) and data structures (Setaluri et al., 2014) can reduce memory footprints but are often computationally expensive and challenging to implement.

We alleviate these limitations by exploring implicit neural representation (Park et al., 2019; Chen & Zhang, 2019; Mescheder et al., 2019) as an alternative *spatial* representation for PDE solvers. Unlike traditional representations that *explicitly* discretize the spatial vector via spatial primitives (e.g., points), neural spatial representations *implicitly* encode the field through neural network weights. In other words, the field is parameterized by a neural network (typically multilayer perceptrons), i.e., $\boldsymbol{f}^n(\boldsymbol{x}) = \boldsymbol{f}_{\theta^n}(\boldsymbol{x})$ with $\theta^n$ being the network weights. As such, the memory usage for storing the spatial field is independent of the number of spatial samples, but rather it is determined by the number of neural network weights. We show that under the same memory constraint, implicit neural representations indeed achieve higher accuracies than traditional discrete representations. Furthermore, implicit neural representations are adaptive by construction (Xie et al., 2021), allocating the network weights to resolve field details at *any* spatial location without changing the network architecture.

Viewed from the lens of optimization-based time integrators, our PDE solver seeks neural network weights that optimize an incremental potential over time (Kane et al., 2000b). Our solver does not employ the so-called training/testing split commonly appearing in many neural-network-based PDE approaches (Sanchez-Gonzalez et al., 2020; Li et al., 2020b). Our approach is the solver itself and does not require training in the machine learning sense. As such, we avoid using the word "training" in the exposition but rather use "optimizing". We employ exactly the same "optimization" integrator formulation as the classical solvers (e.g., finite element method (Bouaziz et al., 2014)).

We compare the proposed solver to grid, mesh, and point cloud representations on time-dependent PDEs from various disciplines, and find that our approach trades wall-clock runtime in favor of three benefits: lower discretization error, lower memory usage, and built-in adaptivity.

## 2 RELATED WORKS

Many prior works have explored representing continuous vector fields with neural networks. Here we highlight two lines of work: implicit neural representation and physics informed neural network.

**Implicit Neural Representation** uses neural networks to parameterize spatially-dependent functions. It has successfully captured the radiance fields (Mildenhall et al., 2020) and the signed distance fields (Park et al., 2019) in computer vision and graphics settings. It has also captured the solutions of strictly spatially dependent PDEs from elastostatics (Zehnder et al., 2021), elliptic PDEs (Chiaramonte et al., 2013), and geometry processing (Yang et al., 2021). Chen et al. (2021), Pan et al. (2022), and Chen et al. (2022) also explore neural networks as spatial representations for dimension reduction. Dupont et al. (2022) develops a machine learning technique operating on data presented as implicit neural representations. Memory consumptions of traditional representations,

such as grids and point clouds, scale poorly with spatial resolutions. Adaptive discretizations can reduce memory but their generations are expensive. By contrast, neural representations are adaptive by construction and can use their representation capacities at arbitrary locations of interest without memory increases or data structures alterations. We refer to the recent review by Xie et al. (2021) for additional contexts.

**Physics Informed Neural Network (PINN)** is a powerful tool for solving PDEs. Traditional numerical methods, such as finite difference and finite element methods, represent tensor fields with well-studied polynomial basis functions (Hughes, 2012) constructed on meshes. Instead, PINN represents tensor fields with neural networks and converts the PDE solution process into finding network weights via PDE-based loss functions. Since the pioneering works by Raissi et al. (2019); Sirignano & Spiliopoulos (2018); Lagaris et al. (1998); Dissanayake & Phan-Thien (1994), PINN has been shown to excellently model forward simulation (Shin et al., 2020; Hennigh et al., 2021; Lu et al., 2021a; Krishnapriyan et al., 2021), inverse design (Raissi et al., 2020; Mao et al., 2020; Mishra & Molinaro, 2022), optimal control (Mowlavi & Nabi, 2021), and uncertain quantification (Lye et al., 2020). PINN has found success in a wide range of application domains, including turbulence (Hennigh et al., 2021), elasticity (Rao et al., 2020), acoustics (Sitzmann et al., 2020), and topology optimization (Zehnder et al., 2021). Due to its mesh-free nature, PINN can robustly handle high-dimensional PDEs. The recent review by Karniadakis et al. (2021) offers more details.

When it comes to using neural networks to represent vector fields, we face an important design choice as which dimension of the vector field is represented through the network. In the case of spatiotemporal vector fields, we have the choice of representing both the spatial and temporal dimensions via neural networks; we can also represent just the spatial dimension or just the temporal dimension with a network.

We observe that the spatial variable $x$ is oftentimes bounded, e.g., a fixed geometry with well-defined boundaries. However, the temporal variable $t$ can be unbounded, e.g., in a virtual reality application where the user interacts with a physical environment indefinitely (Sun et al., 2018). Modeling the additional temporal dimension also puts extra burden on the network. Motivated by these observations, we opt to treat the spatial and the temporal dimensions *differently*. In particular, we use the neural network strictly as a spatial representation and do not consider the temporal dimension as an input to the network. We then evolve this spatial representation by updating the network weights $\theta^n$ (See Figure 2), potentially for an indefinite amount of time. Such an approach is different from standard PINN that takes both the spatial dimension $x$ and the temporal dimension $t$ as an input to the network (Raissi et al., 2019; Karniadakis et al., 2021) which cannot resolve PDE solution outside a pre-defined temporal range (Kim et al., 2021) (See Figure 1).

Du & Zaki (2021); Bruna et al. (2022); Krishnapriyan et al. (2021) also explore evolution of neural network weights over time, with the goal of resolving PINN's limited time range as well as solving high-dimensional problems that classical solvers often suffer. Our work differs from these works by focusing on low-dimensional settings (1D-3D) that heavily rely on classical solvers (e.g., finite element method). Our primary goal is to understand *if we only replace classical solver's spatial representation with a neural network, while keeping the rest unchanged (e.g., time integrator, boundary condition), what tradeoffs do we get?*

**Optimization Time Integrators.** Since our approach only replaces the spatial representations of traditional numerical solvers with neural networks while keeping the rest of the solver intact, it is compatible with *any* classical time integration schemes (e.g., implicit Euler). In particular, we formulate time integration as an energy minimization problem (Radovitzky & Ortiz, 1999; Kane et al., 2000b; Marsden & West, 2001; Kharevych et al., 2006). These integrators find wide applications in PDE solvers on traditional representations, such as grids (Batty et al., 2007), tetrahedral meshes (Bouaziz et al., 2014), and point clouds (Gast et al., 2015). In the case of neural spatial representations, time integration translates into optimizing the neural network weights at every time step.

**Machine Learning for PDEs** is an emerging field with exciting techniques, such as graph neural network (Sanchez-Gonzalez et al., 2020), neural operator (Li et al., 2020b;c), and DeepONet (Lu et al., 2019). These techniques usually train on a dataset and are then validated on a test dataset. However, due to the machine learning nature, these methods's time-stepping schemes neither enforce PDE constraints at test time (Pfaff et al., 2020) nor generalize to scenarios (e.g., initial conditions, boundary conditions) drastically different from the training cases (Wang & Perdikaris, 2021). As a

major point of departure, our approach does *not* employ any training data. There is not a so-called training/inference separation in our approach. Our method is the solver itself, just like the classical solvers (e.g., FEM). As such, we enjoy classical solver's unparalleled generalizability and explicit PDE constraints. See Table 1 for a comparision of these techniques. Relatedly, Wandel et al. (2020) also proposes a data-free approach but still employs a training / testing split.

## 3 Method: Time-stepping on Neural Spatial Representations

Our goal is to solve time-dependent PDEs on neural-network-based spatial representations. In Section 3.1, we first discuss representing spatial vector fields with neural networks. Afterward, we will describe our time-stepping technique that evolves from one neural spatial representation to another.

### 3.1 Neural Networks as Spatial Representations

We parameterize each of the time-discretized spatial vector fields with a neural network: $\boldsymbol{f}^n = \boldsymbol{f}_{\theta^n}$, where $\theta^n$ are the neural network weights at time $t_n$. Specifically, the field quantity at an arbitrary spatial location $\boldsymbol{x} \in \Omega$ can be queried via network inference $\boldsymbol{f}_{\theta^n}(\boldsymbol{x})$.

Traditional representations *explicitly* discretize the spatial vector field using primitives such as points, tetrahedra, or voxels. These primitives *explicitly* correspond to spatial locations due to their compactly supported basis functions (Hughes, 2012). By contrast, neural spatial representations *implicitly* encode the vector field via neural network weights. These weights do not directly correspond to specific spatial locations. Instead, each weight affects the vector field globally. Such global support is also an attribute of spectral methods (Canuto et al., 2007a;b). Compared to spectral methods, our approach does not need to know the required complexity ahead of time in order to determine the ideal basis functions (Xie et al., 2021). Our neural representation automatically optimizes its parameters to where field detail is present.

Whereas memory consumption of traditional *explicit* representations scales poorly with the number of spatial samples, memory consumption for *implicit* neural representations is independent of the number of spatial samples (Xie et al., 2021). Rather, memory use is determined by the number of neural network weights.

**Network Architecture** Following the implicit neural representation literature, we adopt a multilayer perceptron (MLP) architecture with SIREN activation function for its accuracy and quick convergence speed advantages (Sitzmann et al., 2020). Each MLP has a total of $\alpha$ hidden layers, each layer of width $\beta$. The specific choice of these hyper-parameters will be described in Section 4.

**Spatial Gradients** Traditional spatial representations (e.g., the finite element method) compute spatial gradients via basis functions. Higher-order gradients require higher-order basis functions. By contrast, a neural spatial representation is $C^\infty$ by construction. We evaluate their gradients via computation-graph-based auto-differentiation with respect to the input (not the weights).

### 3.2 Temporal Evolution

Given previous-time spatial vector fields $\{\boldsymbol{f}^n(\boldsymbol{x})\}_{k=0}^n$, optimization-based time integrators compute the next time-step ($t_{n+1}$) vector field by optimizing

$$\boldsymbol{f}^{n+1} = \operatorname*{argmin}_{\boldsymbol{f}^{n+1}} \sum_{\boldsymbol{x} \in \mathcal{M} \subset \Omega} \mathcal{I}(\Delta t, \{\boldsymbol{f}^k\}_{k=0}^{n+1}, \{\nabla \boldsymbol{f}^k\}_{k=0}^{n+1}, \{\nabla^2 \boldsymbol{f}^k\}_{k=0}^{n+1}, \ldots) . \tag{2}$$

Traditional time integrators, whether explicit and implicit, can be expressed in optimization forms (Kharevych et al., 2006). Furthermore, this optimization formulation applies to *any* spatial representation, and has been explored thoroughly for traditional discretizations (Batty et al., 2007; Bouaziz et al., 2014; Gast et al., 2015), which is defined over a finite number of the spatial integration samples $\mathcal{M} := \{\boldsymbol{x}^j \in \Omega \mid 1 \leq j \leq |\mathcal{M}|\}$, e.g., grids or meshes. Applying this formulation to a neural spatial representation, we optimize for

$$\theta^{n+1} = \operatorname*{argmin}_{\theta^{n+1}} \sum_{\boldsymbol{x} \in \mathcal{M} \subset \Omega} \mathcal{I}(\Delta t, \{\boldsymbol{f}_{\theta^k}\}_{k=0}^{n+1}, \{\nabla \boldsymbol{f}_{\theta^k}\}_{k=0}^{n+1}, \{\nabla^2 \boldsymbol{f}_{\theta^k}\}_{k=0}^{n+1}, \ldots) \tag{3}$$

where $\{\theta^k\}_{k=0}^n$ are the (fixed, not variable) neural network weights from previous time steps. Figure 2 illustrates our time integration process. The particular choice of the objective function $\mathcal{I}$ depends on the PDE of interest. In all the examples presented in this work, we solve this time-integration optimization problem via Adam (Kingma & Ba, 2014), a first-order stochastic gradient descent method.

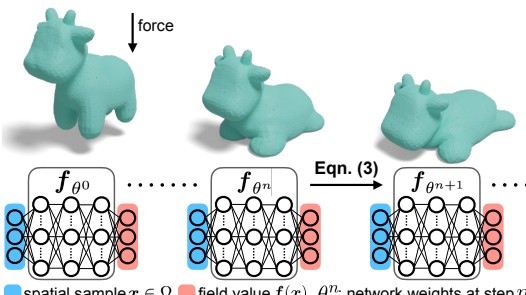

Figure 2: **Time integration.** We represent the field of interest using a neural network $\boldsymbol{f}_{\theta^n}$, whose weights $\theta^n$ are updated at each timestep via an optimization problem (Equation (3)). In this case, the spatial domain $\Omega$ is the interior of the initial object and the represented field $\boldsymbol{f}$ is the deformation map. The governing PDE is the elastodynamics equation (see Section 4.3).

**Spatial Sampling** *Explicit* spatial representations (e.g., tetrahedra mesh) are often tied to a particular spatial sampling; remeshing is sometimes possible, but can also have drawbacks, especially in higher dimensions (Alliez et al., 2002; Narain et al., 2012). By contrast, *implicit* spatial representations allow for arbitrary spatial sampling by construction (Equation (3)).

Following Sitzmann et al. (2020), we dynamically sample $\mathcal{M}$ during optimization. For every gradient descent iteration in every time step, we use a stochastic sample set $\mathcal{M}$ from the spatial domain $\Omega$; $\mathcal{M}$ corresponds to the "mini-batch" in stochastic gradient descent, with batch size $|\mathcal{M}|$. By directly drawing samples from the entire spatial domain $\Omega$, our approach is reminiscent of mesh-free Monte Carlo methods (Sawhney & Crane, 2020).

**Boundary Condition** PDEs are typically accompanied by spatial (e.g., Dirichlet or Neumann) boundary conditions, which we formulate as additional penalty terms in the objective Equation (3),

$$
\begin{aligned}
\theta^{n+1} = \underset{\theta^{n+1}}{\operatorname{argmin}} \sum_{\boldsymbol{x} \in \mathcal{M} \subset \Omega} & \mathcal{I}(\Delta t, \{\boldsymbol{f}_{\theta^k}\}_{k=0}^{n+1}, \{\nabla \boldsymbol{f}_{\theta^k}\}_{k=0}^{n+1}, \{\nabla^2 \boldsymbol{f}_{\theta^k}\}_{k=0}^{n+1}, \ldots) \\
& + \lambda \sum_{\boldsymbol{x}^b \in \mathcal{M}^b \subset \partial \Omega} \mathcal{C}(\boldsymbol{f}_{\theta^{n+1}}, \nabla \boldsymbol{f}_{\theta^{n+1}}, \nabla^2 \boldsymbol{f}_{\theta^{n+1}}, \ldots),
\end{aligned}
\tag{4}
$$

where $\lambda$ is the weighting factor and $\partial \Omega$ is the boundary of the spatial domain. The particular choice of the boundary constraint function $\mathcal{C}$ depends on the problem of interest.

**Initial Condition** The neural network is initialized using the given initial condition, i.e., the field value at time $t = 0$, by optimizing

$$
\theta^0 = \underset{\theta^0}{\operatorname{argmin}} \sum_{\boldsymbol{x} \in \mathcal{M} \subset \Omega} ||\boldsymbol{f}_{\theta^0}(\boldsymbol{x}) - \hat{\boldsymbol{f}}^0(\boldsymbol{x})||_2^2,
\tag{5}
$$

where $\hat{\boldsymbol{f}}^0$ is the given initial condition. Similar to Equation (3), we solve this optimization problem using Adam (Kingma & Ba, 2014) and stochastically sample $\mathcal{M}$ at each gradient descent iteration.

## 4 EXPERIMENTS

In this section, we evaluate our method on three classic time-dependent PDEs: the advection equation, the inviscid Navier-Stokes equation and, the elastodynamics equation. For each problem, we first discuss the continuous PDE and the specific objective function $\mathcal{I}$ for temporal evolution (recall Equation (3)). Then we demonstrate the advantages of our approach by comparing with baselines using discrete spatial representations (i.e. a grid, tetrahedral mesh, or point cloud). We refer readers to Appendices B and C for other implementation details (e.g., initial and boundary conditions) and additional results. The temporal evolutions of the PDEs are best illustrated by the **supplementary video**.

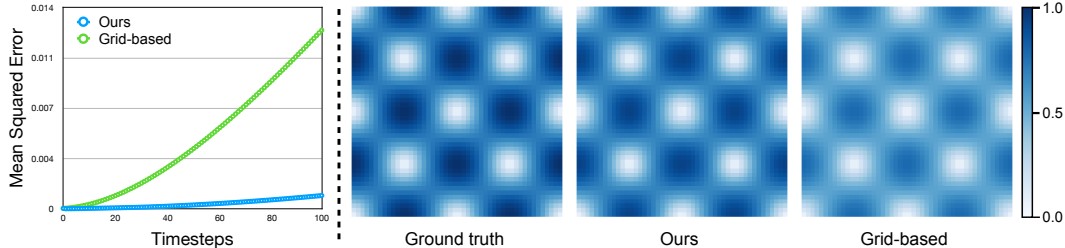

Figure 3: **2D Taylor-Green vortex simulation.** Left: mean squared error of the velocity field for 100 timesteps. Right: velocity magnitude of solutions from the ground truth, ours, and the grid-based method at timestep $n = 100$. Under the same memory usage (for storing the spatial representation), our solution has a significantly smaller error than the grid-based method.

### 4.1 ADVECTION EQUATION

Consider the classic 1D advection equation,

$$\frac{\partial u}{\partial t} + (a \cdot \nabla)u = 0 \,, \tag{6}$$

where $a$ is the advection velocity, and the vector field of interest is the advected quantity $\boldsymbol{f} = u$. It is well known that traditional spatial representations, such as grid-based finite differences, exhibit numerical dissipation for the advection equation (Courant et al., 1952; Selle et al., 2008).

**Time Integration** We adopt the same time integration scheme in both the traditional representation and ours. Choosing the energy-preserving midpoint method (Mullen et al., 2009) yields the time integration operator

$$\boldsymbol{\mathcal{I}} = \|\frac{u^{n+1}(\boldsymbol{x}) - u^n(\boldsymbol{x})}{\Delta t} + (a \cdot \nabla)(\frac{u^{n+1}(\boldsymbol{x}) + u^n(\boldsymbol{x})}{2})\|_2^2 \,. \tag{7}$$

**Results** Figure 1 compares our results with those of grid-based finite differences and PINN (Raissi et al., 2019), subject to equal memory usage of the three methods. A Gaussian-shaped wave moves with constant velocity $a = 0.25$. Our approach uses $\alpha = 2$ hidden layers of width $\beta = 20$, and the finite difference grid resolution is 901. We set PINN to use the same network architecture (with SIREN activation) as ours. For ours and grid-based methods, we set $\Delta t = 0.05$. PINN does not require $\Delta t$ but needs a *pre-specified* temporal range for training. For this temporal range, we use $[0, 3]$. As shown in Figure 1, the solution from the grid-based method diffuses over time due to its spatial discretization. While PINN can accurately capture the result up to $t = 3s$, it fails to produce meaningful solutions beyond its trained temporal range (see $t = 12s$) (Kim et al., 2021). By contrast, our solution does not suffer from numerical dissipation and agrees well with the ground truth at all frames (see the error plot in Figure 1).

### 4.2 INVISCID NAVIER-STOKES EQUATIONS

In the incompressible and inviscid Navier-Stokes Equations

$$\rho_f(\frac{\partial \boldsymbol{u}}{\partial t} + \boldsymbol{u} \cdot \nabla \boldsymbol{u}) = -\nabla p + \rho_f \boldsymbol{g},$$
$$\nabla \cdot \boldsymbol{u} = 0, \tag{8}$$

the vector field of interest is the fluid velocity field $\boldsymbol{f} = \boldsymbol{u}$; $p$ is the pressure, $\boldsymbol{g}$ is the external force, and $\rho_f$ is the fluid density. In our experiments, we consider $\rho_f = 1$ and $\boldsymbol{g} = 0$. The pressure field $p$ is represented with another MLP network.

**Time Integration** We apply the Chorin-style operator splitting scheme (Chorin, 1968; Stam, 1999) to both the neural spatial and finite-difference grid representations. The scheme involves three sequential steps: advection (adv), pressure projection (pro), and velocity correction (cor).

**Advection** uses a semi-Lagrangian method, encoded by the operator (Staniforth & Côté, 1991)

$$\boldsymbol{\mathcal{I}}_{adv} = \|\boldsymbol{u}_{adv}^{n+1}(\boldsymbol{x}) - \boldsymbol{u}^n(\boldsymbol{x}_{backtrack})\|_2^2 \,, \tag{9}$$

whose optimization yields the advected velocity $\boldsymbol{u}_{adv}^{n+1}$. The backtracked location is given by $\boldsymbol{x}_{backtrack} = \boldsymbol{x} - \Delta t \boldsymbol{u}^n(\boldsymbol{x})$. While traditional spatial representations compute the backtracked velocity using interpolation (e.g., linear basis function), our approach *requires no interpolation*, only direct evaluation via network inference at the location $\boldsymbol{x}_{backtrack}$.

**Pressure projection** is encapsulated by the operator

$$\boldsymbol{\mathcal{I}}_{pro} = \|\nabla^2 p^{n+1}(\boldsymbol{x}) - \nabla \cdot \boldsymbol{u}_{adv}^{n+1}(\boldsymbol{x})\|_2^2. \tag{10}$$

Plugging $\boldsymbol{\mathcal{I}}_{pro}$ into the optimization solver, we obtain the pressure $p^{n+1}$ that enforces incompressibility. Note that the MLP that represents the velocity field $\boldsymbol{u}_{adv}$ is kept fixed in this step.

**Velocity correction** is formulated by the operator

$$\boldsymbol{\mathcal{I}}_{cor} = \|\boldsymbol{u}^{n+1} - (\boldsymbol{u}_{adv}^{n+1}(\boldsymbol{x}) - \nabla p^{n+1}(\boldsymbol{x}))\|_2^2 , \tag{11}$$

which adds the pressure gradient to the advected velocity yielding the *incompressible* velocity $\boldsymbol{u}^{n+1}$.

**Results** We first test our method on the 2D Taylor-Green vortex with zero viscosity (Taylor & Green, 1937; Brachet et al., 1983). The closed-form analytical solution is given by: $\boldsymbol{u}(\boldsymbol{x}, t) = (\sin x \cos y, -\cos x \sin y)$ for $\boldsymbol{x} \in [0, 2\pi] \times [0, 2\pi]$. To compare under the same memory usage (for storing the velocity field), we use $\alpha = 3$ hidden layers of width $\beta = 32$ for our MLP and set grid resolution to $48$ for the grid-based projection method. We set $\Delta t = 0.05$ and execute both methods for $100$ timesteps. In Figure 3, we show the mean squared error of the solved velocity field over time. This example demonstrates that our method excellently preserves a stationary solution. Compared to the grid-based method, our method has less diffusion and achieves higher accuracy.

For discrete grid representation, efficiently capturing multi-scale details usually requires difficult-to-implement adaptive data structures (Setaluri et al., 2014). Instead, implicit neural representations are adaptive by construction (Xie et al., 2021) and enable us to capture more details under the same memory storage. We setup an example where the initial velocity field is composed by two Taylor-Green vortices of different scales (see Figure 8 for illustration). We compare our approach with PINN and the grid-based projection method under the same memory constraint for storing the spatial representations. Specifically, our approach and PINN uses a MLP with $\alpha = 3$ hidden layers of width $\beta = 32$ and the grid-based projection method uses resolution $48$. We execute our approach and the grid-based method for $50$ timesteps with $\Delta t = 0.05$, and train PINN with the same temporal range of $2.5$ seconds. Using the solved velocity field, we advect a density field to visualize the amount of fine details captured by different representations. As shown in Figure 4, we are able to capture the fine details of the smaller vortex and best approximate the reference solution. The grid-based method (resolution 48) suffers from severe dissipation and fails to capture the vorticity. PINN is unable to correctly capture this two-vortices field and we found its training loss remains high ($\sim 1e^{-3}$) after convergence. This is in agreement with previous findings (Chuang & Barba, 2022) that suggest PINN approaches have difficulty solving inviscid Navier-Stokes equations for non-trivial examples involving turbulence.

### 4.3 ELASTODYNAMICS EQUATION

In the third experiment, we study the Elastodynamics equations

$$\rho_0 \ddot{\boldsymbol{\phi}} = \nabla \cdot \boldsymbol{P}(\boldsymbol{F}) + \rho_0 \boldsymbol{b} \tag{12}$$

that describe the motions of deformable solids (Gonzalez & Stuart, 2008). The vector field of interest is the deformation map $\boldsymbol{f} = \boldsymbol{\phi}$. Here $\rho_0$ is the density in the reference space, $\boldsymbol{P}$ is the first Piola-Kirchhoff stress, $\boldsymbol{F} = \nabla \phi$ is the deformation gradient, $\dot{\phi}$ and $\ddot{\phi}$ are the velocity and acceleration, and $\boldsymbol{b}$ is the body force.

We assume a hyper-elasticity constitutive law, i.e., $\boldsymbol{P} = \frac{\partial \Psi}{\partial \boldsymbol{F}}$, where $\Psi$ is the energy density function. In particular, we assume a variant of the stable Neo-Hookean energy (Smith et al., 2018)

$$\Psi = \frac{\lambda}{2} \text{tr}^2(\Sigma - \boldsymbol{I}) + \mu(\det(\boldsymbol{F}) - 1)^2, \tag{13}$$

where $\lambda$ and $\mu$ are the first and second lame parameters, $\Sigma$ are the singular values of the deformation gradient $\boldsymbol{F}$, and $\det(\boldsymbol{F})$ is the determinant of the deformation gradient $\boldsymbol{F}$. When $\mu = 0$, the elastic energy recovers the As-Rigid-As-Possible energy (Sorkine & Alexa, 2007).

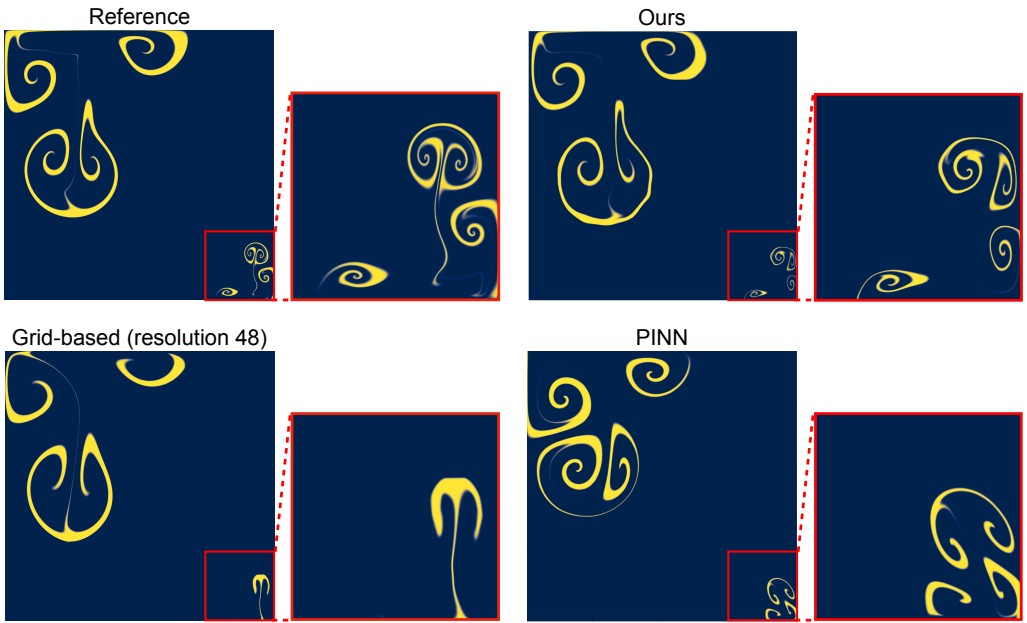

Figure 4: **Two vortices of different scales.** We show the advected density field after 2.5 seconds from the reference (top-left), our method (top-right), the grid-based method of resolution 48 (bottom-left) and PINN (bottom-right). The reference is obtained by running the high-resolution grid-based method (we use resolution 1024). Our MLP ($\alpha = 3$, $\beta = 32$) has the same memory footprint as grids of resolution 48. PINN uses the same MLP network as ours. Under the same memory constraint, our approach suffers from less dissipation, captures more vorticity, and best resembles the reference solution, whose grids takes $\sim 450\times$ memory compared to our network. See Figure 8 for the initial condition of this example.

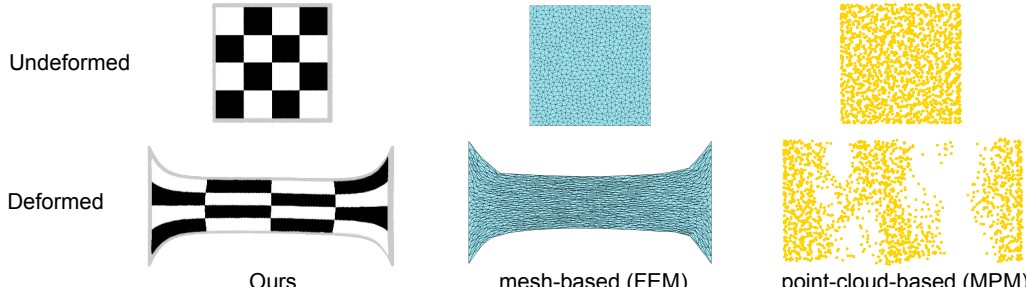

Figure 5: **Elastic tension test**. We compare the quasi-static simulation using our implicit neural representation (left) with the mesh representation (middle), and the point cloud representation (right). Our method handles large deformations matching the mesh-based finite element method (FEM), while the point cloud based material point method (MPM) suffers from incorrect numerical fracture.

**Time Integration** We apply the implicit Euler time integration scheme (Gast et al., 2015; Kane et al., 2000a) to the (1) tetrahedral finite element, (2) material point method, and (3) our neural representation, using the operator

$$\mathcal{I} = \underbrace{\frac{1}{2}\rho_0(\dot{\phi}^{n+1} - \dot{\phi}^n)^T(\dot{\phi}^{n+1} - \dot{\phi}^n)}_{\text{kinematic energy}} + \underbrace{\Psi(\phi^{n+1})}_{\substack{\text{elastic} \\ \text{energy}}} - \underbrace{\rho_0 b^T \phi^{n+1}}_{\substack{\text{external force} \\ \text{potential}}}, \tag{14}$$

where $\dot{\phi}^{n+1} = (\phi^{n+1} - \phi^n)/\Delta t$, $\rho_0$ is the density, $b$ is the external force. We can also incorporate boundary conditions, e.g., positional and contact constraints, by introducing additional energy terms (Bouaziz et al., 2014; Li et al., 2020a) (see Appendix B.4).

**Results** We first compare our implicit neural representation to the traditional tetraheral mesh representation (Finite Element Method, FEM (Hughes, 2012; Reddy, 2019)) and the point cloud rep-

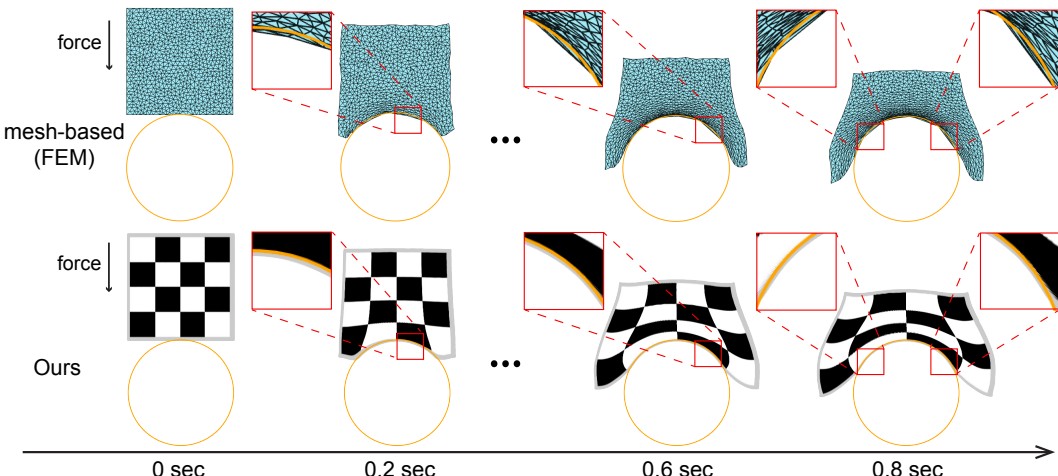

Figure 6: **When an elastic square collides with a circle,** the finite element mesh (top) conforms poorly at the interface compared to the neural representation (bottom), for equal memory usage.

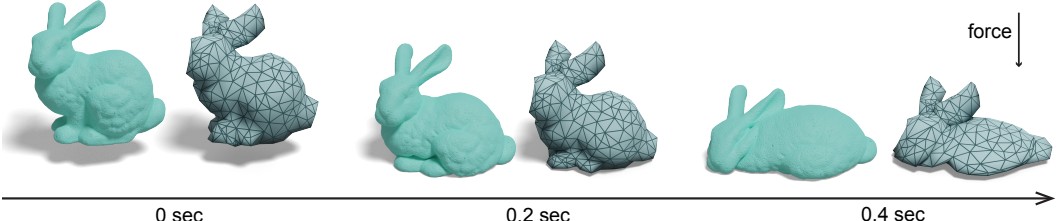

Figure 7: **A bunny collides with the ground in 3D.** Our neural representation (green, left) captures more intricate geometry details and complex dynamics compared to the traditional tetrahedral mesh representation (blue, right) under the same memory usage.

resentation (Material Point Method, MPM (Sulsky et al., 1995; Jiang et al., 2016)). We use $\alpha = 3$ hidden layers of width $\beta = 68$ for our MLP, which takes the same memory as the FEM mesh (0.8K vertices, 1.5K faces) and MPM point cloud (1.7K points). As shown in Figure 5 and Figure 11, our method is capable of handling the large elastic deformations and matches the result of the traditional mesh-based method (FEM), while the point-cloud-based method (MPM) suffers from incorrect numerical fracture due to its meshless nature. To avoid these fractures, meshless methods require sophisticated modifications of the underlying kernel and basis functions (Gray et al., 2001; Su et al., 2022).

By virtue of its *implicit* nature, our representation is able to represent more intricate details compared to the traditional *explicit* representations under the same memory usage. In Figure 6, we show that our implicit neural representation allows the deformed square to gracefully fit the boundary of the sphere during the non-trivial collision. In contrast, the traditional mesh-based representation struggles to produce smooth result due to its insufficient mesh resolution. To alleviate such artifacts, the traditional mesh-based representation either needs to increase resolutions thusly inducing higher memory cost or conducts complex remeshing (Narain et al., 2012). In Figure 7, our implicit neural representation allows for more complex dynamics and fine geometry details compared to the traditional tetrahedral mesh representation.

Note that we adopt the same collision detection and handling strategy for both the neural representation and the mesh-based representation (FEM). Specifically, we use a spring-like penalty force and the corresponding energy to move the collided point out of its collision surface, similar to (McAdams et al., 2011; Xian et al., 2019). Since our approach and FEM share the same time integration scheme and the same collision handling method, the difference reported in Figure 6 and Figure 7 strictly stems from the underlying spatial representations.

These advantages extend to other complex 3D simulations. Figure 2 and Figure 13 depict a cow and statue deforming as they collide with the ground, exhibiting complex geometry, and rich contact-induced deformations.

## 5  DISCUSSION AND CONCLUSION

In this work, we explore *implicit* neural representations as spatial representations for numerically modeling time-dependent PDEs. This representation naturally integrates with widely adopted optimization-based time integrators. PDE solvers with neural spatial representation offers improved accuracy, reduced memory, and automatic adaptivity compared to traditional *explicit* representations such as a mesh, grid, or point cloud.

While offering important benefits, neural-spatial-representation-based PDE time-stepping requires longer wall-clock computation time than existing methods (see also Table 1 by Zehnder et al. (2021) and Section 7 by Yang et al. (2021)). Optimizing neural networks weights takes longer than optimizing grid values even if there are fewer number of neural network weights than the number of grid nodes. For instance, for the bunny example (Figure 7), our neural network optimization takes around 30 minutes per timestep while the corresponding FEM simulation takes less than 1 minute. Future work therefore lies in exploring advanced training techniques that reduce training time (Liu et al., 2020; Martel et al., 2021; Takikawa et al., 2021). In particular, Müller et al. (2022) offer a promising direction where they show that we can reduce implicit neural representation training time from hours to seconds via advanced data structures and optimized implementation.

Our work demonstrates the effectiveness of neural spatial representations in solving time-dependent PDEs and observes empirical convergence under refinement (see Figure 12). Future work should consider theoretical analysis (Mishra & Molinaro, 2022) on convergence and stability. More challenging physical phenomena, such as turbulence (Wilcox et al., 1998), intricate contacts (Johnson & Johnson, 1987), and thin shells (Pfaff et al., 2020), are also important future directions. Currently, our work enforces "soft" boundary conditions. Enforcing "hard" boundary conditions on a neural architecture is another exciting direction (Lu et al., 2021b).

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

## A  COMPARISON OF DIFFERENT PDE SOLVERS

In the table below, we compare other ML-PDE solvers including MeshGraphNet (Pfaff et al. (2020)), GraphNetworkSim (Sanchez-Gonzalez et al. (2020)), DeepOnet (Lu et al. (2019)), Fourier Neural Operator (Li et al. (2020b)).

Table 1: **Comparison with other ML-PDE solvers.**

| Methods | Require training data (from classical solvers or real-world data) | Generalize to out-of-distribution initial/boundary conditions, material parameters, and time spans | Enforce PDE constraints |
|---|---|---|---|
| MeshGraphNet | Yes | Limited | None |
| GraphNetworkSim | Yes | Limited | None |
| DeepOnet | Yes | Limited | Good |
| Fourier Neural Operator | Yes | Limited | Good |
| Ours | No | Good | Good |
| Classical solvers (e.g., FEM) | No | Very good | Very good |

## B  IMPLEMENTATION DETAILS

### B.1  OPTIMIZATION

We solve our time-integration optimization problem (Equation (3)) with the Adam optimizer (Kingma & Ba, 2014). For all examples in our experiments, we set an initial learning rate $\mathbf{lr}_0$ and reduce it by a factor of $0.1$ if the loss value does not decrease for $\mathbf{iter}_p$ iterations. We stop the optimization process when the learning rate is smaller than $\mathbf{lr}_{min}$ or until it reaches a maximum of $\mathbf{iter}_{max}$ iterations. Specific values of these hyper-parameters are described for each example below. We implement our method using PyTorch library and performed our experiments on an NVIDIA GeForce RTX 3090 GPU.

---

**Algorithm 1:** Temporal Evolution

**Input:** network weights $\theta^0$, time step size $\Delta t$, time integrator $\mathcal{I}$, spatial domain $\Omega$

1   $n \leftarrow 0$;
2   **while** *true* **do**
3     $\theta^{n+1} \leftarrow \theta^n$;
4     **while** *not converged* **do**     // network training with Adam optimizer
5       randomly sample $\mathcal{M} \subset \Omega$;
6       $\theta^{n+1} \leftarrow \theta^{n+1} - \alpha \nabla \sum_{\boldsymbol{x} \in \mathcal{M}} \mathcal{I}(\Delta t, \{\boldsymbol{f}_{\theta^k}\}_{k=0}^{n+1}, \{\nabla \boldsymbol{f}_{\theta^k}\}_{k=0}^{n+1}, \ldots)$;     // eq. (3)
7     **end while**
8     $n \leftarrow n + 1$;
9   **end while**

---

### B.2  ADVECTION EQUATION

For our advection example in Figure 1, the 1D spatial domain is $\Omega = [-2, 2]$. We consider the Dirichlet boundary condition, i.e., the advected quantity at boundaries equals zero. Hence we set the boundary constraint term in Equation (4) as

$$\mathcal{C} = ||u^{n+1}(\boldsymbol{x})||_2^2, \tag{15}$$

with the weighting factor $\lambda = 1$. The initial condition for this example is

$$\hat{u}^0(\boldsymbol{x}) = e-\frac{(\boldsymbol{x} - \mu)^2}{2\sigma^2}, \tag{16}$$

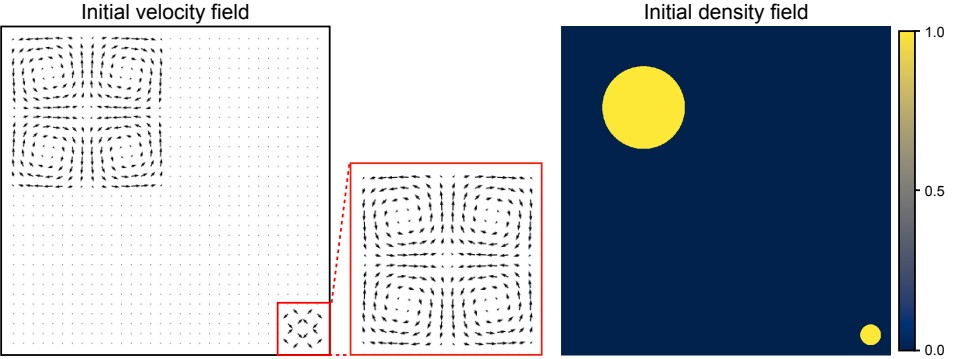

Figure 8: **Initial condition for the example in Figure 4.** Left: velocity field (Equation (19)). Right: density field (Equation (20)).

with $\mu = -1.5$ and $\sigma = 0.1$. We set the optimization hyper-parameters $\mathbf{lr}_0 = 1e-4$, $\mathbf{lr}_{\min} = 1e-8$, $\mathbf{iter}_{\mathrm{p}} = 500$ and $\mathbf{iter}_{\max} = 20000$. For each gradient descent iteration, we randomly sample $|\mathcal{M}| = 5000$ points within the spatial domain $[-2, 2]$. For this example, our method takes $\sim 80s$ to compute per timestep, while the grid-based method (using the same memory) takes $\sim 4e-3s$.

### B.3 INVISCID NAVIER-STOKES EQUATIONS

For our 2D fluid examples, the spatial domain is $\Omega = [-1, 1] \times [-1, 1]$. We consider solid boundary conditions, i.e., the fluid cannot go through the boundaries. Recall that we adopt the operator splitting scheme. Therefore, the boundary constraint terms for the three sequential steps are

$$\mathcal{C}_{adv} = ||\boldsymbol{u}_{adv\perp}^{n+1}(\boldsymbol{x})||_2^2$$
$$\mathcal{C}_{pro} = ||\nabla_\perp p^{n+1}(\boldsymbol{x})||_2^2 \qquad (17)$$
$$\mathcal{C}_{cor} = ||\boldsymbol{u}_\perp^{n+1}(\boldsymbol{x})||_2^2$$

where $\perp$ indicates the perpendicular direction against the boundary. The weighting factor $\lambda = 1$.

**2D Taylor-Green vortex** Standard 2D Taylor-Green is originally defined in domain $[0, 2\pi] \times [0, 2\pi]$. We translate and scale the domain to $[-1, 1] \times [-1, 1]$ such that the input range fits our MLP with the SIREN activation (Sitzmann et al., 2020). Therefore, the initial condition for the velocity field becomes

$$\hat{\boldsymbol{u}}^0(\boldsymbol{x}) = (\frac{1}{\pi}\sin[\pi(x+1)]\cos[\pi(y+1)], -\frac{1}{\pi}\cos[\pi(x+1)]\sin[\pi[y+1]]). \qquad (18)$$

After the simulation, we convert it back to domain $[0, 2\pi] \times [0, 2\pi]$ for evaluation and comparison. We set the optimization hyper-parameters $\mathbf{lr}_0 = 1e-5$, $\mathbf{lr}_{\min} = 1e-8$, $\mathbf{iter}_{\mathrm{p}} = 500$ and $\mathbf{iter}_{\max} = 20000$. The size of sample set $|\mathcal{M}| = 256^2$. For this example, our method takes $\sim 10\mathrm{min}$ to compute per timestep, while the grid-based method (using the same memory) takes $\sim 0.03s$.

**Two vortices of different scale** For the example shown in Figure 4, the initial condition for the velocity field is

$$\hat{\boldsymbol{u}}^0(\boldsymbol{x}) = \begin{cases} (\sin[2\pi(x+1)]\cos[2\pi(y+1)], -\cos[2\pi(x+1)]\sin[2\pi(y+1)]) & \boldsymbol{x} \in [-1, 0]^2 \\ (\sin[8\pi(x-\frac{7}{4})]\cos[8\pi(y-\frac{7}{4})], -\cos[8\pi(x-\frac{7}{4})]\sin[8\pi(y-\frac{7}{4})]) & \boldsymbol{x} \in [\frac{7}{4}, 1]^2 \\ (0, 0) & \text{otherwise.} \end{cases} \qquad (19)$$

The density field that we advect is initialized as

$$\hat{d}^0(\boldsymbol{x}) = \begin{cases} 1 & ||2\boldsymbol{x} + 1|| \leq 0.5 \text{ or } ||8\boldsymbol{x} + 7|| \leq 0.5 \\ 0 & \text{otherwise.} \end{cases} \qquad (20)$$

Figure 8 visually illustrates the above initial conditions. After the simulation, we convert it back to domain $[0, 2\pi] \times [0, 2\pi]$ for evaluation and comparison. We set the optimization hyper-parameters $\mathbf{lr}_0 = 1e-5$, $\mathbf{lr}_{\min} = 1e-8$, $\mathbf{iter}_{\mathrm{p}} = 500$ and $\mathbf{iter}_{\max} = 20000$. The size of sample set $|\mathcal{M}| = 128^2$. For this example, our method takes $\sim 10\mathrm{min}$ to compute per timestep.

Table 2: **Experiment setup for the elasticity examples.** $\rho_0$ is the density. $\lambda$ and $\mu$ are the first and second lame parameters. $\alpha$ and $\beta$ are the number of hidden layers and the dimension of the hidden features. $\mathbf{lr}_0$ and $\mathbf{iter}_{\max}$ are the initial learning rate and the maximum number of iterations. $\mathbf{t}_{\text{avg}}$ is average training time per time step. Note that the density $\rho_0$ and timestep size **dt** are reported as N/A for the quasistatic example Stretch (2D) (Figure 5).

| Example | Dim | $|\mathcal{M}|$ | dt | $\rho_0$ | $\lambda$ | $\mu$ | $\alpha$ | $\beta$ | $\mathbf{lr}_0$ | $\mathbf{iter}_{\max}$ | $\mathbf{t}_{\text{avg}}(s)$ |
|---|---|---|---|---|---|---|---|---|---|---|---|
| **Collision (2D) (Figure 6)** | 2 | $100^2$ | 0.1 | $1e1$ | $2e1$ | $1e3$ | 3 | 68 | $1e-5$ | $1e4$ | $1.38e2$ |
| **Stretch (2D) (Figure 5)** | 2 | $100^2$ | N/A | N/A | $1e0$ | $1e3$ | 3 | 68 | $1e-4$ | $5e4$ | $2.30e3$ |
| **Bunny (Figure 7)** | 3 | $20^3$ | 0.1 | $1e0$ | $1e2$ | $1e3$ | 3 | 66 | $1e-5$ | $2e4$ | $1.70e3$ |
| **Spot (Figure 2)** | 3 | $20^3$ | 0.1 | $1e0$ | $1e2$ | $1e3$ | 3 | 66 | $1e-3$ | $5e3$ | $1.74e3$ |
| **Lucy (Figure 13)** | 3 | $20^3$ | 0.1 | $1e0$ | $1e3$ | $1e3$ | 3 | 128 | $1e-4$ | $2e4$ | $1.16e3$ |

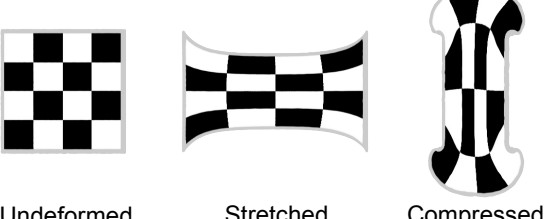

Undeformed     Stretched     Compressed

Figure 9: **Elasticity patch test.** Quasistatic simulation in 2D (undeformed, stretched, compressed).

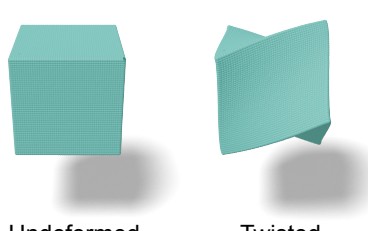

Undeformed     Twisted

Figure 10: **Twisting test.** Quasistatic simulation in 3D. The right end is twisted 45 degrees.

### B.4 Elastodynamics Equation

**Initial and Boundary Conditions** For our 2D elasticity examples in Figure 5, Figure 9 and Figure 6, the 2D spatial domain is $\Omega = [-1, 1] \times [-1, 1]$. For our 3D elasticity examples in Figure 12, the 3D spatial domain is $\Omega = [-1, 1] \times [-1, 1] \times [-1, 1]$. For our 2D and 3D examples involving nonregular geometry (Figure 7, Figure 2 and Figure 13), the spatial domain is the interior of the shape, including the boundary. The initial condition for all the elasticity examples is

$$\hat{\phi}^0(\boldsymbol{x}) = (0, 0) \text{ (2D)},$$
$$\hat{\phi}^0(\boldsymbol{x}) = (0, 0, 0) \text{ (3D)} \tag{21}$$

The boundary constraint for elasticity examples involves positional constraints or collision constraints. Positional constraints, or *Dirichlet boundary conditions*, can be realized by defining the position of the constraint set $\partial\Omega$ as the desired goal positions $\overline{\phi}_{\partial\Omega}$:

$$\mathcal{I}_{\text{pos}} = \|\phi_{\partial\Omega}^{n+1} - \overline{\phi}_{\partial\Omega}\|_2^2. \tag{22}$$

Collision constraints can be handled by adding unilateral constraints dynamically and viewing the collision penalty force as external force. Specifically, for a colliding point $\boldsymbol{q}_c$, we first find the closest surface point $\boldsymbol{b}_c$ with normal $\boldsymbol{n}_c$, and define our spring-like collision penalty force as:

$$\boldsymbol{f}_{\text{col}} = k_{\text{col}}((\boldsymbol{b}_c - \boldsymbol{q}_c)^\top \boldsymbol{n}_c)\boldsymbol{n}_c. \tag{23}$$

where $k_{\text{col}}$ is the ratio for the collision penalty force.

The corresponding collision energy can be defined as the work exerted by the collision force:

$$\mathcal{I}_{\text{col}} = \rho_0 \boldsymbol{f}_{\text{col}}^T \phi^{n+1}. \tag{24}$$

**Experiment Setup** For all the 2D comparison under the same memory usage, we use $\alpha = 3$ hidden layers of width $\beta = 68$ with SIREN activation function (Sitzmann et al., 2020) for our MLP, which takes the same memory (57 KB) as the FEM mesh (0.8K vertices, 1.5K faces) and MPM point cloud (1.7K points) in use. We initialize the 2D deformation field of the network to be zero

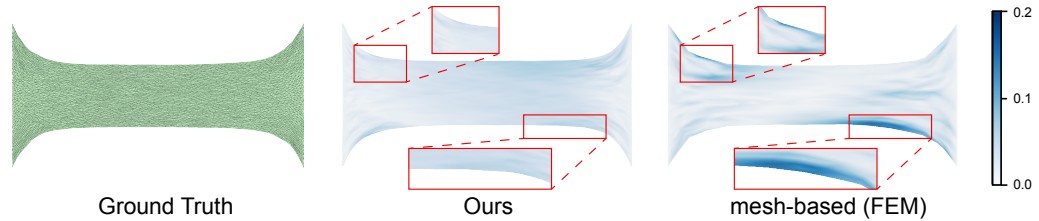

Figure 11: **Error of the elastic tension test.** We visualize the L2 position error **e** of our result and the low-resolution FEM result (0.8K vertices) in Fig. 5 with respect to high-resolution FEM ground truth (3.2K vertices). Under the same memory footprint (for storing the spatial representation), our method ($\|\mathbf{e}\|_\infty$ = 8.89e-2) is closer to the high-resolution ground truth than the mesh-based method ($\|\mathbf{e}\|_\infty$ = 1.99e-1).

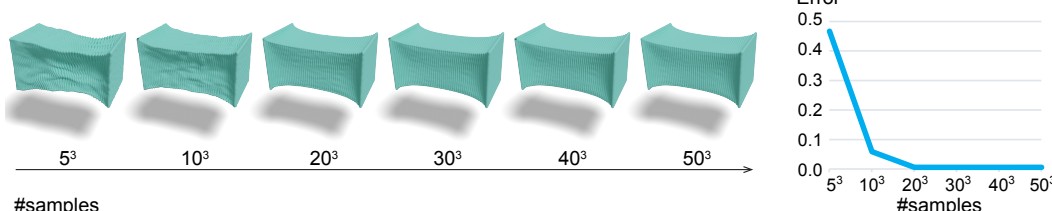

Figure 12: **Sampling convergence test**. Quasi-static elasticity simulation using different number of samples. We trained the implicit neural representation using different number of samples and visualized the result using samples $|\mathcal{M}| = 50^3$ (left). We further compute the error with respect to $|\mathcal{M}| = 50^3$ when using different number of samples (right). As we increase the number of the training samples in use, the deformation field converges to the result trained on the highest resolution.

using $|\mathcal{M}| = 1000^2$ uniform and random samples. Then we train the network using $|\mathcal{M}| = 100^2$ uniform and random samples at each training iteration. We use Bartels (Levin, 2020) and Taichi (Hu et al., 2019) to perform the FEM and MPM simulation, respectively. We run our FEM and MPM comparison on CPU using a MacBook Pro with Apple M2 processor and 24GB of RAM.

For the 3D comparison under the same memory usage, for the bunny example (Figure 7), we use $\alpha = 3$ hidden layers of width $\beta = 66$ with SIREN activation function for our MLP, which takes the same memory (53 KB) as the FEM mesh (0.5K vertices, 1.5K tetrahedra) in use. For the statue example (Figure 13), we use $\alpha = 3$ hidden layers of width $\beta = 128$ with SIREN activation function for our MLP, which takes the same memory (197 KB) as the FEM mesh (2.0K vertices, 7.0K tetrahedra) in use. We initialize the 3D deformation field of the network to be zero using $|\mathcal{M}| = 100^3$ uniform and random samples. Then we train the network using $|\mathcal{M}| = 20^3$ uniform and random samples at each training iteration. Here for simplicity we use the mesh vertices as the uniform samples. We further report all the parameters and experiment setup in Table 2. In addition, we set the hyper-parameters $\mathbf{iter}_\mathrm{p} = 800$ and $\mathbf{lr}_\mathrm{min} = 1e-8$ for all elasticity examples.

For sampling of the shapes involving nonregular geometry, for simplicity we choose to use a triangle or tetrahedral mesh and perform sampling within it. An ideal alternative would be adopting the implicit representation of the surface and performing rejection sampling based on it.

For rendering, we simply sample sufficient number of points from the undeformed shape and evaluate the trained model at time $t$ on the sample positions to predict their deformation. Here we only sample the surface of the shape in 3D cases. Then we render the shape as a dense point cloud.

## C   ADDITIONAL RESULTS

### C.1   ELASTODYNAMICS EQUATION

We validate the physical plausibility of our method using a small 2D patch test (Figure 9) and a 3D twisting example (Figure 10). We show that our implicit neural representation can exhibit volume-preserving property under both stretching, compression and twisting.

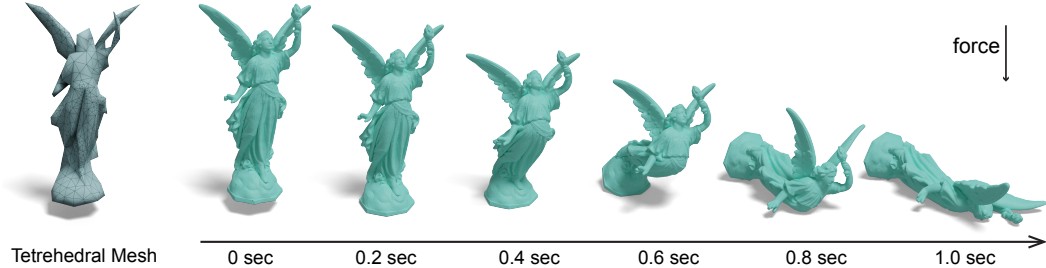

Figure 13: **The statue collides with the ground and deforms.** Our implicit neural representation (green, right) is capable of capturing more fine geometry details compared to the traditional tetrahedral mesh representation (blue, left) under the same memory footprint.

We demonstrate *qualitative* and *quatitative* convergence of our method when increasing the number of training samples in use. In Figure 12, we compare the quasi-static stretching results visualized using $|\mathcal{M}| = 50^3$ uniform samples when using different number of training samples ($|\mathcal{M}| = 5^3, 10^3, 20^3, 30^3, 40^3, 50^3$), and report the error with respect to the high-resolution trained result (trained and visualized both on $|\mathcal{M}| = 50^3$).

Finally, we provide an additional example for elasticity involving complex contact-induced deformations in Figure 13. Our implicit neural representation is able to maintain more intricate geometry details compared to the traditional tetrahedral mesh representation under the same memory usage.

## D  QUANTITATIVE RESULTS

We present the quantitative results (error, runtime and memory usage) for each of our tested examples in separate tables.

Table 3: **Quantitative results for 1D advection example (Figure 1).** Error: mean absolute error over total 240 time steps, compared to the ground truth analytical solution. Time: runtime for total 240 time steps. Memory: memory usage for storing the spatial representations.

| Methods | Error | Time | Memory |
|---------|-------|------|--------|
| Ours | 0.0030 | 5.33h | 3.520KB |
| Grid | 0.0146 | 1.13s | 3.520KB |
| PINN | 0.0564 | 9.73m | 3.598KB |

Table 4: **Quantitative results for 2D Taylor-Green fluid example (Figure 3).** Error: mean squared error of velocity field over total 100 time steps, compared to the ground truth analytical solution. Time: runtime for total 100 time steps. Memory: memory usage for storing the spatial representations.

| Methods | Error | Time | Memory |
|---------|-------|------|--------|
| Ours | 3.35e-4 | 14.02h | 25.887KB |
| Grid-based | 4.83e-3 | 2.91s | 27.00KB |

Table 5: **Quantitative results for two-vortices fluid example (Figure 4).** Error: mean absolute error of kinetic energy over total 50 time steps, compared to the reference solution (obtained by high resolution grid-based method). Kinetic energy is computed using $1024^2$ uniform samples. Time: runtime for total 50 time steps. Memory: memory usage for storing the spatial representations.

| Methods | Error | Time | Memory |
|---------|-------|------|--------|
| Ours | 2.24e2 | 10.81h | 25.887KB |
| Grid-based | 1.07e4 | 1.78s | 27.00KB |
| PINN | 1.92e4 | 2.2h | 26.137KB |
| PINN-seq | 3.21e4 | 20.83h | 26.137KB |

Table 6: **Quantitative results for 2d tension example (Figure 5 and Figure 11).** Error: Infinity norm of L2 distance w.r.t. high-resolution ground truth. Time: Total runtime until convergence. Memory: memory usage for storing the spatial representations.

| Methods | Error | Time | Memory |
|---|---|---|---|
| Ours | 8.82e-2 | 38.33m | 56.32KB |
| Mesh-based | 1.99e-1 | 22.04s | 54.00KB |

Table 7: **Elastic square collides with circle (Figure 6).** Error: Infinity norm of L2 intersection distance w.r.t. the circle boundary. Time: Average runtime for 1 time step. Memory: memory usage for storing the spatial representations.

| Methods | Error | | | | Time | Memory |
|---|---|---|---|---|---|---|
| | 0.2s | 0.4s | 0.6s | 0.8s | | |
| Ours | 1.62e-2 | 1.10e-2 | 1.04e-2 | 1.06e-2 | 2.30m | 56.32KB |
| Mesh-based | 3.45e-2 | 1.39e-2 | 5.47e-2 | 4.23e-2 | 9.82s | 54.00KB |

