# OpenReview forum: "Implicit Neural Spatial Representations for Time-dependent PDEs"
_ICLR.cc/2023/Conference — Submitted to ICLR 2023_

### Official Review · Reviewer_Tkaf · 2022-10-17

**Confidence:** 4
**Correctness:** 4
**Technical Novelty And Significance:** 3
**Empirical Novelty And Significance:** 3
**Recommendation:** 6

**Clarity, Quality, Novelty And Reproducibility:**

The paper is written very clearly and is very high quality. While the method is heavily based on prior work using SIRENs, there is some technical novelty and the paper shows that it leads to increased results (some more could be done to further drive this point home). The paper seems reproducible.

**Strength And Weaknesses:**

In my opinion, the main strengths of the paper are:
- The paper is written clearly, and communicates the proposed method well. The claims made in the paper are well supported by the experiments, and are well explained. The relationship of the proposed method to the related work, grid based methods and PINNs, is clearly outlined so the contribution and differences proposed are extremely clear.
- The method seems to result in noticeable qualitative improvements in the solutions to various PDEs. The evaluation is thorough, and is done across three different PDEs demonstrating that the result is not just overfit for a particular PDE.

In my opinion, the main weaknesses of the paper are:
- The amount of novelty is slightly limited. It seems like the method builds on the SIREN architecture heavily, and the only main contribution is the separation of the temporal dimension and treating it differently than the spatial dimensions. Do the PINN network baselines also use the SIREN architecture, and then just treat the time dimension as an input to the network? If not, this baseline absolutely needs to be compared to in detail in order to prove that this contribution of separating time and spatial dimensions is critical for performance. Overall, I would think the paper should make more clear that this is the main contribution over the prior state-of-the-art in using coordinate-based networks for PDE solutions.
- There are very limited quantitative comparisons. While the qualitative comparisons are very nice and do show that there is a difference in performance, it’s difficult for me to understand the level of the failure without some kind of quantitative metric for each of the experiments. Addition of some quantitative results would definitely increase the strength of the paper by demonstrating that the method proposed really improves on the performance of the prior work.


**Summary Of The Paper:**

The paper proposes a method for partial differential equations by using a coordinate-based network instead of a standard grid as the representation for the solution to the equation. This is done by modeling the spatial dimensions as a SIREN MLP, and then evolving the parameters of the network over the time dimension by solving an optimization problem at each time step. This explicit separation allows the solution to better generalize to new times, as the coordinate-based network tends to overfit on bounded signals. The paper demonstrates that this works for a variety of partial differential equations, and that despite the slower speed, the approach outperforms grid-based solvers in terms of accuracy and memory consumption.

**Summary Of The Review:**

Overall, I find the paper to be good: it very clearly proposes and describes a novel tweak to the existing SIREN architecture for PDE solutions (treating the temporal and spatial dimensions differently), evaluates this over a wide variety of different PDEs, and seems to demonstrate that the method performs significantly better. Since the technical contribution is based very heavily on the previous work, I think that the main key for the paper should be to demonstrate that this tweak does significantly improve the results. The paper does do this, but it could definitely be made stronger with some more explanation of what the baselines are (is PINN a SIREN), and some quantitative results to demonstrate more improved performance alongside the qualitative results.

POST REBUTTAL UPDATE:
After reading the author response to my and other reviews, I am inclined to maintain my score. I appreciate the additional comparisons added in the Appendix which highlight the increase in performance (and differences in runtime, which I didn't recognize at first look of the manuscript without these tables). This improves my opinion of the work. However, I still believe that the amount of technical contribution is limited since the method is very similar to PINN with SIREN, and only offers increased performance at the expense of additional compute time. Overall, I still think it is a good paper that could be accepted and is above the threshold.

---

> ### Author Response · Authors · 2022-11-13
> **Official Comment by Paper943 Authors**
>
> Thank you for your thoughtful review of our work and wonderful feedback!
>
> - Q: Do the PINN network baselines also use the SIREN architecture, and then just treat the time dimension as an input to the network?
> - A: Yes, the PINN baselines we compared use the same SIREN architecture as ours. The only difference is that PINN additionally treats the time dimension as an input. We clarified this point in the revised version (see Sec. 4.1 “Results” paragraph).
>
>
>
> - Q: There are very limited quantitative comparisons.
>
> - A: We add tables of quantitative results for each example and Figure 11 in Appendix D.

---

> > ### Comment · Reviewer_Tkaf · 2022-12-02
> > **Response**
> >
> > Thank you for the response to my questions and concerns! I appreciate the addition of the extra quantitative contributions, and have updated my review. Thanks.

---

### Official Review · Reviewer_8oNn · 2022-10-22

**Confidence:** 4
**Correctness:** 3
**Technical Novelty And Significance:** 2
**Empirical Novelty And Significance:** 2
**Recommendation:** 5

**Clarity, Quality, Novelty And Reproducibility:**

Clarity: The idea is clearly exposed in the paper.

Quality: The proposed model is tested on different equations, but with limited comparison w.r.t existing neural approaches and numerical solvers.

Novelty: Somehow novel with respect to PINNs.

Reproducibility: Hyperparameters and architecture details are given in Appendices. However, no source code is given.

**Strength And Weaknesses:**

Strength:
- The approach uses coordinate neural networks that can be evaluated at any location in space, which is more flexible than commonly used solvers defined on grid or meshes.
- The optimization is more reasonable than the original PINNs for time-dependent PDEs: the residual loss based on time-stepping introduces the causal temporal relation into the optimization.

Weaknesses:
- As the paper proposes a new meshless PDE solver, it is then necessary to mention the background of the solvers of the same type. Also, the comparison to the existing meshless PDE solvers other than PINNs is somehow limited. If other meshless PDE solvers turn out to be as effective as the proposed method, can the claim about the trade-off between wall-clock runtime and other advantages still holds?
- Comparison w.r.t approches proposed for the same objective of going beyond the training temporal horizon of PINNs is needed, e.g. (Wang & Perdikaris, 2021).
- Question about the comparison with the PINNs in Section 4.2 inside the same training horizon: are PINNs trained as in the original paper (Raissi et al., 2019)? How the PINNs perform with sequence-like training compared to the proposed method (e.g. Krishnapriyan et al., 2021), which is closer to the setting of the paper?


References:
- (Raissi et al., 2019) Physics-informed neural networks: A deep learning framework for solving forward and inverse problems involving nonlinear partial differential equations
- (Krishnapriyan et al., 2021) Characterizing possible failure modes in physics-informed neural networks
- (Wang & Perdikaris, 2021) Long-time integration of parametric evolution equations with physics-informed DeepONets

**Summary Of The Paper:**

This paper proposes to use spatial-coordinate neural networks as an alternative to the spatial representations used in traditional solvers for time-dependent partial differential equations. This work reformulates the PINN-like PDE solvers in a time-dependent way. It calculates the spatial derivatives as in PINNs at each time step and optimizes an INR w.r.t a physical loss involving a discretized time derivative at each time step. The proposed approach extends effectively PINNs beyond the limited temporal domain.

**Summary Of The Review:**

This paper introduces the time-stepping into the original PINNs framework, which appears to outperform PINNs and some traditional solvers in many use cases. However, due to the limited novelty and comparison with the baseline methods, I think the paper is marginally below the acceptance threshold.

---

> ### Author Response · Authors · 2022-11-13
> **Official Comment by Paper943 Authors**
>
> Thank you for your valuable review and feedback! We really appreciate it.
>
> - Q: Discussion and comparison to existing meshless PDE solvers other than PINN.
>
> - A:
>
>   - In Figure 5, we demonstrate a comparison with the meshless material point method (MPM). Due to its meshless nature, it suffers from unphysical, numerical fracture. The research community has observed similar errors with the meshless smoothed-particle hydrodynamics (SPH) method.
>   - In general, mesh-free Galerkin, RBF, and MLS still require assignments of DoFs to spatial domains. As such adaptivity remains challenging \[1] and memory consumption scales linearly with the number of spatial samples. By contrast, our approach supports straightforward remeshing and the memory consumption is determined by the number of neural network weights, not the spatial samples.
>   - Meshless spectral methods assign DoFs to frequency domains instead of spatial domains. These frequency domain basis functions must be chosen ahead of time. Once chosen, the method cannot capture other frequencies. By contrast, our approach does not need to know the required complexity ahead of time in order to determine the ideal basis functions \[2]. Our neural representation automatically optimizes its parameters to where field detail is present.
>
>
>
> - Q: Comparison w.r.t approaches proposed for the same objective of going beyond the training temporal horizon of PINNs is needed, e.g. (Wang & Perdikaris, 2021).
>
> - A:
>
>   - We agree with the reviewer that there are many great works extending beyond the training range (see the revised intro and related work).
>
>   - As a major point of departure from these approaches, including (Wang & Perdikaris, 2021), our approach does NOT use any training data. Our approach does not employ the so-called training/inference separation. Our method is the solver itself, just like the classical solvers (e.g., FEM). Our approach adopts virtually the same implementation as any classical solver formulated with optimization-time integrators \[3]. As such, we enjoy classical solver's unparalleled generalizability and explicit PDE constraints. By contrast, as discussed in (Wang & Perdikaris, 2021, end of section 3), "the trained model may not generalize very well for out-of-distribution initial conditions."
>
>   - We would like to take this opportunity to emphasize that the primary goal of our work is not extending PINN beyond the training temporal horizon. This is one of the many consequences of our optimization time integrator formulation. Our primary goal is to understand "if we ONLY replace classical solver's spatial representation with a neural network, while keeping the rest unchanged (e.g., time integrator, boundary condition), what tradeoffs do we get?" In this work, we discover that a neural-spatial-representation-based solver offers a unique advantage of accuracy, memory, and adaptivity at the cost of longer computing time (under the current standard neural network training framework).
>
>
>
> - Q: Are PINNs trained as in the original paper (Raissi et al., 2019)?
>
> - A: Yes, we follow the original paper to train PINN, and use the same SIREN architecture as ours for fair comparison.
>
>
>
> - Q: How do PINNs perform with sequence-like training compared to the proposed method (e.g. Krishnapriyan et al., 2021), which is closer to the setting of the paper?
>
> - A: Since their official git repo (<https://github.com/a1k12/characterizing-pinns-failure-modes>) doesn’t include the sequential training code, we implement this sequential training scheme by ourselves and compare with it on our two-vortices example. The quantitative result is added to Table 5 in the appendix. Our result is still better than this PINN variant.
>
>      We hypothesize the performance difference mainly comes from different optimization objectives. With explicit temporal discretization, our method naturally supports different time integrators. For the Navier-Stokes equations, we are able to apply the Chorin-style operator splitting scheme, which translates a nonlinear PDE into three linear PDEs. Consequently, it is easier to optimize than the original N-S equations. This PINN variant with sequential training scheme, still treats the continuous space-time domain as a whole. Therefore, it is unable to use the operator splitting scheme and needs to directly optimize the residual of the nonlinear N-S equation. Essentially, by formulating time integrators the same way as classical methods, we are able to build upon the rich literature of classical PDE solvers.
>
>
>
> \[1] Gao+ An adaptive generalized interpolation material point method for simulating elastoplastic materials, SIGGRAPH ASIA 2017
>
> \[2] Xie+ Neural fields in visual computing and beyond, CGF 2022
>
> \[3] Bouaziz+ Projective dynamics: Fusing constraint projections for fast simulation. SIGGRAPH 2014

---

> > ### Comment · Reviewer_8oNn · 2022-11-25
> > **Post rebuttal response**
> >
> > Thank you for your response. I appreciate the discussion on meshless solvers.
> > However, I am still concerned about the lack of comparison because the proposed method is doing the same task --- solving PDEs with given physical conditions and residual --- as (Wang & Perdikaris, 2021) and the method itself is also very similar to it. The arguments that rule out the comparison are somewhat weak for me. A humble motivation cannot dismiss the high similarity to other methods.
> > Also, contrary to your comment, (Wang & Perdikaris, 2021) works in solver mode as your method. The approach have only a loss on the initial/boundary condition and given physical residual loss (see the loss definition in Eq. (3.3) and other concrete examples in Section 4). No supervision on simulated trajectories is needed. This method works even without discretizing the temporal operator inside the $\Delta t$. With single initial condition, the setting is the same as yours. They have a training set of "initial conditions" as they try to solve simultaneously multiple trajectories. So I still think it is necessary to discuss more cautiously the differences with (Wang & Perdikaris, 2021) and compare with it. Therefore, I am leaning towards maintaining my rating.

---

> > > ### Author Response · Authors · 2022-12-01
> > > **Re: Comparison to (Wang & Perdikaris, 2021)**
> > >
> > > We thank the reviewer for the patient response and detailed suggestions!
> > >
> > > We agree with the reviewer that (Wang & Perdikaris, 2021) does not require training data from classical solvers or real experiments, thanks to their physics-informed loss. Indeed, it can be used for the solver mode. We have since downloaded their source code https://github.com/PredictiveIntelligenceLab/Long-time-Integration-PI-DeepONets and compared it against our method.
> > >
> > > Before detailing the comparison result, we would like to point out that all experiments in (Wang &  Perdikaris, 2021) train on multiple initial conditions and run network inference for time evolution. However, as suggested by the reviewer, we use it in solver mode by training on a single initial condition so that it can be compared with the similar setups presented in our paper.
> > >
> > > Below is the quantitative result on the 2D multi-scale fluid vorticities example (see figure 4 of our paper).
> > > | Methods                               | Error        | Time   | Memory |
> > > |---------------------------------------|--------------|--------|-------------|
> > > | Ours                                  | 2243.664737  | 10.81h | 25.887KB    |
> > > | Grid - 48                             | 10677.294127 | 1.78s  | 27.00KB     |
> > > | PINN                                  | 19206.759164 | 2.2h   | 26.137KB    |
> > > | (Wang & Perdikaris, 2021) solver mode | 19282.506586 | 13.33h | 81.324KB    |
> > > Error: average kinetic energy difference over 50 timesteps. Time: total run time for 50 timesteps. Memory: memory footprint to store the representation.
> > >
> > > Our solution has a significantly smaller error than their solution. For a fair comparison, we use the same network structure as ours for the branch net and trunk net in (Wang & Perdikaris, 2021). Specifically, 3 hidden layers of width 32 and SIREN activation. We empirically found that using SIREN leads to faster and better convergence than tanh activation they used in their paper. In addition, we also tried increasing the network capacity but didn’t see significant improvement of their method.
> > >
> > > Similar to our comparison with (Krishnapriyan et al., 2021) (see our rebuttal response above), we hypothesize the performance difference mainly comes from different optimization objectives. With clear temporal discretization, our method naturally supports different time integrators. For the Navier-Stokes equations, we are able to apply the Chorin-style operator splitting scheme, which translates a nonlinear PDE into three linear PDEs. Consequently, it is easier to optimize than the original N-S equations. Though (Wang & Perdikaris, 2021) can work in solver mode, it still treats the continuous space-time domain as a whole, just like PINN. Therefore, it is unable to use the operator splitting scheme and needs to directly optimize the residual of the highly nonlinear N-S equation. Essentially, by formulating time integrators the same way as classical methods, we are able to build upon the rich literature of classical PDE solvers.
> > >
> > > Relatedly, we are able to obtain similar results using (Wang & Perdikaris, 2021) and our method on the 1D advection example (see Figure 1 of our paper). 1D advection is relatively simple and we indeed share a very similar optimization objective. However, as mentioned in the previous paragraphs, our method outperforms their approach on the more involved N-S equation with multi-scale vorticities, thanks to the robust time integrators that we employ.
> > >
> > > We thank the reviewer once again. Please don’t hesitate to let us know should the reviewer think that there are any other relevant comparisons.

---

### Official Review · Reviewer_uDb6 · 2022-10-29

**Confidence:** 4
**Correctness:** 3
**Technical Novelty And Significance:** 3
**Empirical Novelty And Significance:** 2
**Recommendation:** 6

**Clarity, Quality, Novelty And Reproducibility:**

The paper is very clear, well structured and easy to follow. The work is relatively original as it is a combination of different existing works but very important for the ML&Physics community.

**Strength And Weaknesses:**

The proposed method is straightforward either from modeling or implementation standpoints. It allows to encode the spatial information through a neural network where its weights store  informations of downstream tasks  in an implicit manner.  In contrary to classical  numerical discretization scheme where their success is highly related to the quality of mesh including the spatial sampling and the complexity of some local regions especially near the boundary layer,  implicit neural representation is independent of the number of points and spatial location since the learned weights are global and as a result affect the vector field globally. Moreover, implicit neural representations are powerful to mitigates the spectral biais of neural networks by leveraging high frequencies information and capturing well fine-grained information without scarifying the generalization capabilities of neural networks. The proposed work is also interesting if we look at the tedious task of numerical solver to build adaptive representation by considering different sampling (remeshing) strategies where their choice highly depends on the task, implicit neural representation are adaptive by construction w.r.t weights of the neural network, hence it is agnostic to the neural network architecture.
Another interesting aspect of this work is spatial gradient computation achieved through auto-differentiation. In traditional methods such VFM, FEM, higher order gradients requires higher order basis function such as Chebyshev polynomial which is very expensive in practice (complexity grows with the targeted order to achieve the desired accuracy). In the proposed work higher order gradients are cheap to get as neural implicit representations can achieve sufficiently high order easily by construction thanks to periodic function.

One of the drawbacks of the proposed work is in the computational time. The authors are aware of it and give some directions to tackle that in future works.

The claims are corroborated with experimental study including different physical systems such as Advection equation, N-S equations, and Elastodynamics equations. The results are in general satisfactory.

I have some considerations and questions:

1/ In N-S experiments, what is the Mach and Reynolds number ?

2/ Have you tested or thought about SIREN variants including Multi-Scale implicit neural representations ? The latter could be more representative since Physical systems are multi-scale by construction.

3/ Implicit Neural Representation are based on periodic functions. I am wondering if the good results you obtain on physical systems described with periodic boundary condition is intrinsically related to this property of periodicity of the neural network and boundary conditions. Have you any results on the same physical systems but described with non-periodic boundary conditions ?

4/  It would be interesting to have more quantitative results and comparisons with recent PINN variants.

5/ One other experiment that would be important to conduct is to assess the generalization capabilities of the proposed model on complex geometries (especially near the boundary layer).  For instance, considering the datasets and tasks described in "Learning mesh-based simulation with graph networks".



**Summary Of The Paper:**

In this work,  a novel design of a time-dependent PDEs solver is proposed. It is achieved via a neural network and based on recent works on implicit neural representations (MLP + SIREN). This method is an alternative to a a well-known family of spatial discretization methods.

**Summary Of The Review:**

The overall work is satisfactory. However,  the experimental part could be drastically consolidated with more challenging tasks and comparison with existing works.

---

> ### Author Response · Authors · 2022-11-13
> **Official Comment by Paper943 Authors**
>
> Thank you for your valuable feedback and kind words!
>
> - Q: In N-S experiments, what is the Mach and Reynolds number?
> - A: As our experiments study incompressible and inviscid flow, the Mach number M$\rightarrow$0 and the Reynolds number R=+$\infty$.
>
>
>
> - Q: Have you tested or thought about SIREN variants including Multi-Scale implicit neural representations? The latter could be more representative since Physical systems are multi-scale by construction.
> - A: We have tried the multi-scale hybrid implicit neural representation \[1], but didn’t find any significant improvement compared to SIREN. We also considered using multi-scale hash-grid encoding \[2], but found its implementation doesn’t support second-order derivatives. Therefore, we are unable to test it on our examples. We opt to leave the usage of multi-scale neural implicit for future work.
>
>
>
> - Q: Do you have any results on the same physical systems but described with non-periodic boundary conditions?
> - A: All examples that we tested use non-periodic boundary conditions. Please see Eqn.(15), Eqn.(17) and Eqn.(22) in the appendix for the exact definition for our used boundary constraints.
>
>
>
> - Q: It would be interesting to have more quantitative results and comparisons with recent PINN variants.
> - A: We add tables of quantitative results in Appendix D. For comparison with PINN variants, we consider the sequential training scheme by \[3], as pointed out by reviewer _8oNn_. On our two vortices fluid example, our approach obtains smaller error than this PINN variant (See Table 5).
>
>
>
> - Q: The ability to handle complex geometries?
>
> - A:
>
>   - We agree with the reviewer that our method can benefit from handling more complex cases. Our code currently does not support thin shell (clothes) models shown in MeshGraphNet. We have since listed it as a future work.
>   - That said, our sphere elasticity collision example (Figure 6, and also 7,10,13) bears similar complexity as the sphere thin plate example demonstrated by the meshGraphNet work (See Figure 2b in their paper). If the reviewer has any specific 3D elasticity example in mind, we would be happy to know!
>
>
>
> \[1] (Takikawa et al., 2021) Neural geometric level of detail: Real-time rendering with implicit 3d shapes
>
> \[2] (Muller et al., 2022) Instant neural graphics primitives with a multiresolution hash encoding.
>
> \[3] (Krishnapriyan et al., 2021) Characterizing possible failure modes in physics-informed neural networks

---

### Official Review · Reviewer_DwBW · 2022-10-31

**Confidence:** 3
**Correctness:** 3
**Technical Novelty And Significance:** 2
**Empirical Novelty And Significance:** Not applicable
**Recommendation:** 5

**Clarity, Quality, Novelty And Reproducibility:**


Clarity:

1.  What are the numerical evaluation metric for solution. Do you compare the generalization error?

2.  I was confused about the comparison of the methods. For example,  In 1D example, the hyperparameters for each method are chosen based on equal memory usage, I don't know the meaning of the 'equal memory usage'. It seems to me that in NN solver, one can choose 5000 points per time step while the finite difference method only has 901 points.  It would be great if the author can make tables displaying time complexity, sample complexity for each example and accuracy for each example so it has clear indication of accuracy and efficiency.



**Strength And Weaknesses:**


Strengths:

The studied problem is interesting and important. The paper is well organized and easy to follow.  The numerical experiments have enough details for reproducibility. The visualizations  of the solution and mesh are nicely presented.


Weakness:

1. Limited technical novelty.  As far as I understand, the contribution of this work is to use neural network to represent the solution, and allows the flexibility of spatial samplings per time step. I would say both ideas are not new and has appeared in many works, and  this paper did not provide further insights on  how to select the sampling set and the design of neural network that yield better results.

 As authors mentioned: one could representing both the spatial and temporal dimensions via neural networks;  or just the spatial dimension or just the temporal dimension. It seems to me that consider spatial dimension may ignore the temporal correlation among different snapshots of the solution, and need more weights that what is actually need to represent a smooth spatiotemproal field.


2. The method is not  thoroughly evaluated. The evaluation metric is vague: there is no clear definition on  memory usage, numerical evaluation error metric for solutions and time complexity for each method.  Did the author use the state of the art solver for each example? How about the same type of PDEs but with different parameters and initial conditions. These PDEs may not allow explicit solutions but the classical solvers have theoretical guarantees to obtain the solution up to certain accuracy, so I think it is fine to compare with them to see if the proposed method have advantages in other aspects.

3. The paper should discuss more on the limitations. For example,  this solver is only for smooth solutions, how to deal with non-smooth case? Also, compared to classical solvers, it  may be more difficult to incorporate certain physical constraints and boundary constraints.

4. The PDE examples are up to 3D where the classical solvers also generally worked well.  It is believed that NN solver has advantages on deal with higher dimension problem (see ref 1) , can the authors comment on the potential of the proposed method on higher dimension problem?

References:
[1] Solving high-dimensional partial differential equations using deep learning



**Summary Of The Paper:**


This paper proposes an NN solver for the time-dependent PDE. The proposed method uses neural networks to parametrize the time-discretized spatial vector field. The loss function for training combined problem-dependent time integrator, penalty terms by boundary condition, and initial condition, which is the same with the classical solvers, yet one is allowed to choose different spatial samplings per time step.  The method is shown to work on several test problems, ranging from 1D advection function to 3D  Elastodynamics equation.   Overall, the authors show that the proposed method is promising for improving the flexibility and numerical accuracy of the solutions.


**Summary Of The Review:**

To summarize, the paper proposed a NN solver that utilize neural network to parametrize the time-discretized spatial vector field and the adaptive meshes per time step.  I would recommend authors to present a more comprehensive and informative comparison with the existing methods, and more insights on why the proposed method has advantages and its possible limitations.

---

> ### Author Response · Authors · 2022-11-13
> **Official Comment by Paper943 Authors (1/2)**
>
> Thank you for your detailed review and valuable feedback!
>
> - Q: Limited technical novelty.
>
> - A:
>
>   - We agree with the reviewer that there are many great neural network based approaches with meshless sampling. We added additional citations and discussions to put our work into perspective (see intro and related work).
>
>   - As a major point of departure from many neural-network-based PDE approaches \[1]\[2], our approach does NOT use any training data. Our approach does not use the so-called training/inference separation. Our method is the solver itself, just like the classical solvers (e.g., FEM). Our approach adopts virtually the same implementation as any classical solver formulated with optimization-time integrators \[3]. The only difference is the spatial representation.
>
>   - \[4,5,6] also explored the evolution of neural network weights over time as a tool for resolving PINN's limited time range as well as solving high-dimensional problems that classical solvers often suffer. Our work differs from those works by focusing on low-dimensional settings (1D-3D) that heavily rely on classical solvers (e.g., finite element method). We discover that, even on these low-dimensional PDEs, neural-network-based solvers offer a unique advantage of accuracy, memory, and adaptivity at the cost of computing time (under the current standard SGD framework).
>
>
>
> - Q: The method is not thoroughly evaluated. The evaluation metric is vague: there is no clear definition on memory usage, numerical evaluation error metric for solutions and time complexity for each method.
>
> - A:
>
>   - To better describe the evaluation metrics, we added a table of quantitative results comparing our method and the baseline. Please see Appendix D in the revised paper.
>
>   - By memory usage, we mean the memory usage for storing the spatial representation. For discrete grid-based or mesh-based methods, it is the amount of memory for storing such grids or meshes.  For our method and PINN, it is the amount of memory for storing neural network weights.
>
>
>
> - Q: Did the author use the state-of-the-art solver for each example?
>
> - A:
>
>   - In every example, to ensure fair comparisons, we use **the same optimization time integrator** formulation for both the baselines and our approach. The ONLY difference in the solver is the spatial representation (under the same memory).
>   - For advection, this optimization time integrator formulation is equivalent to the classical midpoint approach, which is industry standard \[8].
>   - For fluids, this optimization time integrator formulation is equivalent to the classical Chorin-style operator splitting scheme, which is also an industry standard \[7].
>   - For elasticity, this optimization time integrator formulation is widely adopted in graphics and robotics \[3]
>
>
>
> - Q: The paper should discuss more on the limitations. For example, this solver is only for smooth solutions, how to deal with non-smooth cases?
>
> - A:
>
>   - Our approach is able to capture the non-smooth high-vorticities flow cases (see Figure 4), in which PINN is known to suffer \[9]
>   - That said, we agree with the reviewer that our approach has difficulty handling non-smooth cases featuring strong discontinuity, such as shock waves. We have since added that as a limitation. To alleviate this issue, future work may consider advanced neural network structure (as opposed to the vanilla multilayer perceptron explored in this work). \[10]

---

> > ### Author Response · Authors · 2022-11-13
> > **Official Comment by Paper943 Authors (2/2)**
> >
> > - Q: Can the authors comment on the potential of the proposed method on higher dimension problems?
> >
> > - A:
> >
> >   - Yes, our approach is fully compatible with higher-dimension problems. In particular, The concurrent work by \[5] adopts a similar architecture but focuses on higher-dimensional problems.
> >   - That said, we would like to slightly push back on the point that “the classical solvers also generally worked well with low-dimensional problems”.
> >   - The explicit spatial discretizations employed by classical solvers are intuitive to model and understand, but these representations are not necessarily optimal for accuracy, memory usage, or adaptivity.
> >   - The core discovery of this work is that even on these low-dimensional PDEs, our neural spatial representation offers some unique advantages: e.g., significantly more accurate under the same memory constraint; adaptively capture solutions of different feature sizes.
> >
> >
> >
> > \[1] Sanchez-Gonzalez+ Learning to simulate complex physics with graph networks, ICML 2020
> >
> > \[2] Li+ Fourier Neural Operator for Parametric Partial Differential Equations, ICLR 2020
> >
> > \[3] Bouaziz+ Projective dynamics: Fusing constraint projections for fast simulation. SIGGRAPH 2014
> >
> > \[4] Du+ Evolutional deep neural network, Physical Review E 2021
> >
> > \[5] Bruna+ Neural Galerkin Scheme with Active Learning for High-Dimensional Evolution Equations, arXiv 2022
> >
> > \[6] Krishnapriyan+ Characterizing possible failure modes in physics-informed neural networks, Neurips 2021
> >
> > \[7] Fedkiw+ Visual simulation of smoke, SIGGRAPH 2001
> >
> > \[8] Mullen+ Energy-preserving integrators for fluid animation, SIGGRAPH 2009
> >
> > \[9] Chuang+ Experience report of physics-informed neural networks in fluid simulations: pitfalls and frustration, arxiv 2022
> >
> > \[10] Müller+ Instant neural graphics primitives with a multiresolution hash encoding, SIGGRAPH 2022

---

> > > ### Comment · Reviewer_DwBW · 2022-11-26
> > > **Post-rebuttal response**
> > >
> > > Thank the authors for their detailed response and I appreciate the authors that adding comparison tables for the experiments and discussion with more refs. In light of this, I would raise my score to 5. However.  the current description and empirical comparisons can not fully convince me the significant advantages over classical solvers in low dimensional settings: it is short of a principled way to guide the users to find ``optimal" representations and it is very time expensive to run a single experiment. I would be more  impressed if the authors can show examples of higher dimensional setting where classical solvers failed.

---

> > > > ### Author Response · Authors · 2022-12-11
> > > > **Re: Post-rebuttal response**
> > > >
> > > > We thank the reviewer for raising the score and for the valuable suggestions!
> > > >
> > > > Indeed, our work focuses on demonstrating the unique features of implicit neural representation on low-dimensional PDEs over traditional spatial representations: accuracy, memory usage, and adaptivity.
> > > >
> > > > For the reviewer’s concern with computation time, recent implicit neural representation works offer promising results (e.g., [1][2]), reducing training time from hours to seconds. We also want to point out that other neural-network-based solvers (e.g., PINN, see Table 1 of [3]) also suffer from long computation time compared to classical solvers when solving forward problems.
> > > >
> > > > We agree that high-dimensional PDE is an exciting application. The concurrent work [4] has indeed explored using similar architecture as ours (neural network as spatial representation) but rather focuses on high-dimensional PDEs. In particular, they show that advanced importance sampling strategies are required to capture the details of high-dimensional problems in a memory-efficient way. In theory, we can also apply their sampling approach to our method. However, we believe this is slightly beyond the scope of our work which focuses on low-dimensional PDEs and have left this as future works.
> > > >
> > > > [1] Muller+ Instant neural graphics primitives with a multiresolution hash encoding. SIGGRAPH 2022 best paper
> > > >
> > > > [2] Takikawa+ Neural geometric level of detail: Real-time rendering with implicit 3d shapes. CVPR 2021
> > > >
> > > > [3] Zehnder+ Ntopo: Mesh-free topology optimization using implicit neural representations. Neurips 2021
> > > >
> > > > [4] Bruna+ Neural Galerkin Scheme with Active Learning for High-Dimensional Evolution Equations. arXiv 2022

---

### Official Review · Reviewer_7x1Y · 2022-11-03

**Confidence:** 3
**Correctness:** 3
**Technical Novelty And Significance:** 4
**Empirical Novelty And Significance:** 4
**Recommendation:** 8

**Clarity, Quality, Novelty And Reproducibility:**

The paper is well-written and easy to follow. The only issue I mentioned above concerns the steps of the inference procedure.

The paper is also of decent quality when we exclude the Navier-Stokes vs Euler Equations.

In terms of novelty, I find the idea very promising as I explained in the first paragraph above. And it seems to be the first time to do dynamical updates over implicit representations.

In my current understanding, reproducibility means that the code is provided in the supplementary material (SM), which is not the case here. In the SM I only found visualizations of the benchmarks. Thus, I would say that the results are not reproducible because there is always some hyperparameter that is not in the paper, even if it is just a random seed.

**Strength And Weaknesses:**

The idea of using low-dimensional implicit spatial representations as a PDE domain discretization is really interesting because it starts with the assumption that there exists a low-dimensional manifold, on which the physical system lives. I could imagine using such representations to study the structure of chaotic systems such as turbulent flows, see Dynamic Mode Decomposition.

Looking at this paper as a first step in the right direction (I have many ideas about how one could improve this method, but this paper is already a legitimate first step), there are some things that can be improved:
1. Reading the paper for the first time, I didn't get how the inference is carried out up until Section 5, which clarifies that the implicit model is retrained at every time step, which of course is a time intense operation. Could you add an algorithm or just a step-by-step description of the inference step in Section 3, or at least in the appendix?
2. In a pure computer vision paper, I could accept demonstrating performance only through visuals or graphs, but in a PDE paper, I would expect a table with results that other papers can compare against. Could you please add such data (even if it is in the appendix)?
3.  Section 4.2 shows the performance of the method on the "Navier-Stokes Equations" (NSE). Having a solid background in fluid mechanics, I have to say that the *inviscid* NSE are called "Euler Equations" and have quite different behavior than the general NSE. Thus, it is for the least misleading to call Section 4.2 "NSE".
4. In the same line of thought as 3., because the Euler Equations have zero viscosity (=> there is no dissipation), the 2D Taylor-Green system has a stationary solution, i.e. nothing will change at all. This is a nice example as a demonstration that the proposed method doesn't add too many errors as would happen if we use any finite spatial discretization, but this should be stated somewhere. I mean something like "This example demonstrates that our method quite well preserves a stationary solution, but it does not yet show any dynamic behavior. For a dynamical 2D simulation we have the example with the two vortices.". Pointing out the preservation of the stationary solution is indeed a strong argument compared to classical methods, which very often entail the so-called numerical diffusion term.

**Summary Of The Paper:**

The paper proposes using implicit neural representations as a domain discretization, on which a PDE acts. For temporal evolution, classical PDE integrators are used. Next to requiring less memory for the representations, the model has the capacity to represent small but relevant features accurately. However, in the current implementation, the inference is more than one order of magnitude slower compared to using a classical PDE solver.

**Summary Of The Review:**

The paper is very insightful, but with some minor details to mend before publication. I would already give the paper a 6, but if my comments are addressed appropriately, it would become an 8.

---

> ### Author Response · Authors · 2022-11-13
> **Official Comment by Paper943 Authors**
>
> Thank you for your careful review of our work and excellent feedback!
>
> - Q: Could you add an algorithm or a step-by-step description of the inference step in Section 3?
>
> - A: We added the pseudocode for our algorithm in Appendix B.1. We’d like to point out there is not a so-called “inference” step, since our method does NOT have training/testing separation. Our network is the solver itself, just like the classical PDE solver. Our approach does not require training in the machine learning sense. As such, we avoid using the word "training" in the exposition but rather use "optimizing" since we employ exactly the same optimization integrator formulation as the classical solvers (e.g., finite element method \[1]). We have clarified in the introduction that in this work “optimization” is our preferred word for “training”.
>
>
>
>
> - Q: I would expect a table with results that other papers can compare against. Could you please add such data?
>
> - A: We added tables of quantitative results in Appendix D.
>
>
>
>
> - Q: _inviscid_ NSE are called "Euler Equations" and have quite different behavior than the general NSE. It is for the least misleading to call Section 4.2 "NSE".
>
> - A: Thanks for the suggestion! We have since updated the section title.
>
>
>
>
> - Q: The zero-viscosity 2D Taylor-Green system is a nice example as a demonstration that the proposed method doesn't add too many errors as would happen if we use any finite spatial discretization, but this should be stated somewhere.
> - A: Thanks for this suggestion. We further emphasize this point in our revised paper (see Sec. 4.2 “Results” paragraph).
>
> \[1] Bouaziz+ Projective dynamics: Fusing constraint projections for fast simulation. SIGGRAPH 2014

---

> > ### Comment · Reviewer_7x1Y · 2022-12-09
> > **response**
> >
> > Thanks for the updates! The additions to the appendix are very valuable.
> > As promised, I updated my recommendation and I'm sorry for the late reply from my side.

---

### Official Review · Reviewer_oNuL · 2022-11-04

**Confidence:** 4
**Correctness:** 3
**Technical Novelty And Significance:** 2
**Empirical Novelty And Significance:** 2
**Recommendation:** 5

**Clarity, Quality, Novelty And Reproducibility:**

While the exposition is written with a lot of clarity, and does have a very high quality at first glance, it displays a severe lack of nuance in its relation to current work to properly assess its originality. While the present paper still has novelty in aspects, the novel arises from the implicit spatial representations, which are also viewed from the viewpoint of implicit neural representations. Previous approaches such as Graph Network Simulations, and MeshGraphNets have already followed a highly similar approach, where one can surmise that under certain conditions the Graph Network Simulations approach of spatial graph network representations, can be equivalent to the presented spatial implicit representations. As such some of the claims such as "To our best knowledge, computing neural spatial representations on time-dependent PDEs for long horizon tasks with multiple time steps has not been explored, and our work aims to fill this gap." are not supported in literature.

There do in addition exist a multitude of sentence structure errors, and typos which I would dearly recommend to address.

**Strength And Weaknesses:**

Strengths:
- Clarity of the mathematical exposition of the approach
- Clarity in the embedding of the presented approaches intersection and background in PDE- and fluid dynamics theory

Weaknesses:
- Poor integration into the current state-of-the-art by omitting many similar approaches, such as Graph Network Simulations [2] which effectively encode the spatial representation with graph neural networks, and similarly use classical time-integrators for the temporal representation. The same applies for the even more modern MeshGraph [3], and Fourier Neural Operator [4], both would need to be compared to the herein presented approach to properly assess the capabilities of the presented approach. The usage of implicit representations for functions has also been explored by Dupont et al. [1] before. Said work would need to be put in context to the presented literature.
- Improperly chosen comparisons to current state-of-the-art models due to an incomplete representation of the literature, see preceding point
- No ablation analyses
- Performance of the approach induces a 30x overhead, hence making it intractable for any practical problem at the current moment

[1] Dupont, Emilien, Hyunjik Kim, SM Ali Eslami, Danilo Jimenez Rezende, and Dan Rosenbaum. "From data to functa: Your data point is a function and you can treat it like one." In International Conference on Machine Learning, pp. 5694-5725. PMLR, 2022.
[2] Pfaff, Tobias, Meire Fortunato, Alvaro Sanchez-Gonzalez, and Peter Battaglia. "Learning Mesh-Based Simulation with Graph Networks." In International Conference on Learning Representations. 2020.
[3] Sanchez-Gonzalez, Alvaro, Jonathan Godwin, Tobias Pfaff, Rex Ying, Jure Leskovec, and Peter Battaglia. "Learning to simulate complex physics with graph networks." In International Conference on Machine Learning, pp. 8459-8468. PMLR, 2020.
[4] Li, Zongyi, Nikola Borislavov Kovachki, Kamyar Azizzadenesheli, Kaushik Bhattacharya, Andrew Stuart, and Anima Anandkumar. "Fourier Neural Operator for Parametric Partial Differential Equations." In International Conference on Learning Representations. 2020.



**Summary Of The Paper:**

In this work the authors propose to implicit neural representations to model the spatial representation, while approaching the temporal representation with classical optimization-based approaches to model time-dependent partial differential equations.

**Summary Of The Review:**

This work present a new approach for implicit neural spatial representations for time-dependent partial differential equations. With the temporal representation modeled classically, there exist a number of similar approaches in existing literature, hence rendering the use of implicit representations the main novelty of the paper. In addition the paper severely lacks adequately chosen comparisons, only comparing to the original physics-informed neural network (PINN) approach from 2019.

---

> ### Author Response · Authors · 2022-11-13
> **Official Comment by Paper943 Authors (1/2)**
>
> Thank you for your thoughtful review and feedback! We really appreciate it.
>
> - Q: Comparison with other ML-PDE approaches \[2]\[3]\[4]
>
> - A:
>
>   - ML-PDE techniques like \[2]\[3]\[4] are typically trained on a dataset and are validated on a test dataset. These datasets are often generated by **classic solvers** (e.g., finite element method). However, due to the machine learning nature, these methods’ time-stepping schemes neither **enforce the PDE** at test time nor **generalize** to out-of-distributions scenarios. As a major point of departure, our approach **does NOT employ any training data**. There isn’t a so-called training/inference separation in our approach. We only replace **classical solvers**’ discrete representation with a neural network. As such, our method explicitly **enforces the PDE ** and is **generalizable** to any initial/boundary conditions, material parameters, and time spans. We have further clarified this in the text (see intro and related work).
>
>   - In the table below, we compare different PDE solvers:
>
>    | Methods                       | Require training data (from classical solvers or real-world data) | Generalize to out-of-distribution initial/boundary conditions, material parameters, and time spans | Enforce PDE |
>     | ----------------------------- | ------------------------------------------------------------ | ------------------------------------------------------------ | ----------- |
>     | MeshGraphNet                  | Yes                                                          | Limited                                                      | None        |
>     | GraphNetworkSim               | Yes                                                          | Limited                                                      | None        |
>     | DeepOnet                      | Yes                                                          | Limited                                                      | Good        |
>     | Fourier Neural Operator       | Yes                                                          | Limited                                                      | Good        |
>     | Ours                          | No                                                           | Good                                                         | Good        |
>     | Classical solvers (e.g., FEM) | No                                                           | Very good                                                    | Very good   |
>
>   - \[1] does not address PDE problems but instead develops an interesting machine learning framework for data presented as implicit neural representations. We have since added it to the related work section.
>
>   - We have removed the sentence starting with “To our best knowledge” to avoid any overclaims and put our work into the perspective of other ML-PDE works (see related work).
>
>
>
> - Q: ablation study
>
> - A:
>
>   - Since our method ONLY replaces classical solvers’ discrete representation with a simple MLP network while keeping the rest of the solver intact, we don’t have many network design choices that need ablation studies.
>
>   - We think the most important design choice is that we only use the network to represent the spatial dimensions and update the network weights by explicit temporal integration. This is a key design choice that differentiates us from PINN and its variants, which jointly represent the spatial and temporal dimensions. From this perspective, the comparison between our method and PINN is an ablation study for this design choice. We have such comparisons on the advection equation (see Figure 1) and N-S equations (see Figure 4). The results show that our method is not limited by a pre-specified temporal range and achieves more accurate solutions. Therefore, we believe our such design choice does have its advantages.
>
>   - If there are any specific ablation studies that the reviewer hopes to see, we’d be happy to include them!

---

> > ### Author Response · Authors · 2022-11-13
> > **Official Comment by Paper943 Authors (2/2)**
> >
> > - Q: impractical timing
> > - A:
> >
> >   - Yes, we acknowledge that the timing in its current state is slower than traditional solvers. This is also the case with other variants of PINN methods. (see also Table 1 by \[5] and Section 7 by \[6]). Optimizing neural network weights takes longer than optimizing grid values even if there are fewer numbers of neural network weights than the number of grid nodes.
> >   - That said, many deep learning techniques were intractably slow, but created an opportunity for novel improvements in optimization methods.
> >   - Furthermore, recent ML research offers promising directions for speeding up implicit neural representation training. In particular, \[7] offers a promising direction where they show that we can reduce implicit neural representation training time from hours to seconds via advanced data structures and optimized implementation.
> >
> > \[1] Dupont+ From data to functa: Your data point is a function and you can treat it like one, ICML 2022
> >
> > \[2] Pfaff+ Learning Mesh-Based Simulation with Graph Networks, ICLR 2020
> >
> > \[3] Sanchez-Gonzalez+ Learning to simulate complex physics with graph networks, ICML 2020
> >
> > \[4] Li+ Fourier Neural Operator for Parametric Partial Differential Equations, ICLR 2020
> >
> > \[5] NTopo: Mesh-free Topology Optimization using Implicit Neural Representations, NeurIPS 2021
> >
> > \[6] Yang+ Geometry Processing with Neural Fields, NeurIPS 2021
> >
> > \[7] Müller+ Instant neural graphics primitives with a multiresolution hash encoding, SIGGRAPH 2022

---

> > > ### Comment · Reviewer_oNuL · 2022-11-18
> > > **Addressing of Concerns & Raising of Score**
> > >
> > > I thank the authors for addressing many of my concerns, especially the added nuance with the relation to the existing body of literature is much appreciated and I believe the authors have addressed many of my concerns in that regard. In light of this I will raise my score.
> > >
> > > For my score to be raised much higher, I would urge the authors to add further experiments comparing to more existing approaches in literature such as FNO, MeshGraphNet, or the below mentioned approach by Wang & Perdikaris. While all of these require training data, a PINN such as the one used in the current comparison also requires training data.

---

> > > > ### Author Response · Authors · 2022-11-18
> > > > **Re: Addressing of Concerns & Raising of Score**
> > > >
> > > > We appreciate the reviewer going through our rebuttal! We find your perspective very valuable.
> > > >
> > > > - We would like to gently point out that PINN does NOT require any **training data from classic solvers or real capture**. For example, \[1] uses PINN to solve poisson equation w/o training data from classic solvers. \[2] uses PINN to solve topology optimization w/o training data from classic solvers; \[3] uses PINN to tackle optimal control w/o training data from classic solvers. Specifically, in our comparison, we implement PINN by following the tutorial \[4] provided by the original PINN author, Maziar Raissi. This tutorial also does not require training data from classic solvers but uses the PDE residual as the loss function evaluated on samples drawn from the spatialtemporal domain. With that said, PINN does also support working with training data from classic solvers.
> > > >
> > > > - Unlike solver-type-approaches (PINN, our approach, and the classical solvers), training-data-dependent (i.e., data from classical solvers) approaches (FNO, MeshGraphNet, and \[Wang & Perdikaris]) are severely limited by their ability to generalize to out-of-distribution initial/boundary conditions, material parameters, and time spans.
> > > >
> > > >   - For example, once trained on a specific thin shell Young's modulus, MeshGraphNet can only model dynamics of this particular Young's modulus but not others.
> > > >   - As another example, \[Wang & Perdikaris] discussed how their approach cannot handle initial conditions beyond the one appearing in the training data (see section 3 last sentence).
> > > >
> > > > - By contrast, solver-type-approaches (PINN, our approach, and the classical solvers) do not have these limitations. Our approach uses an explicit PDE model and we can easily swap in with different material parameters and initial conditions.
> > > >
> > > > ****
> > > >
> > > > \[1] Chiaramonte+ Solving differential equations using neural networks, arXiv
> > > >
> > > > 2013
> > > >
> > > > \[2] Zehnder+ NTopo: Mesh-free Topology Optimization using Implicit Neural Representations. NeurIPS 2021
> > > >
> > > > \[3] Mowlavi+ Optimal control of PDEs using physics-informed neural networks, Journal Computational Physics 2022
> > > >
> > > > \[4] Raissi+  <https://maziarraissi.github.io/PINNs/>

---

> > > > > ### Comment · Reviewer_oNuL · 2022-11-19
> > > > > **PINNs**
> > > > >
> > > > > I would politely disagree, or at the very least mark this statement as "misleading" - PINNs require a number of collocation points within the domain, and at the boundary condition. Exact details depend on the type of PINN used. E.g. the residual points are usually randomly spaced within the domain. As such you do indeed utilize training pairs.

---

> > > > > > ### Author Response · Authors · 2022-11-21
> > > > > > **Re: PINNS**
> > > > > >
> > > > > > - We thank the reviewer for the patient response. We really appreciate your constructive feedback.
> > > > > >
> > > > > >   - We would like to take this opportunity clarify what we mean by “training data”.
> > > > > >   - The reviewer is indeed right that PINN needs to be trained, but the “training data” it needs are simply a set of sampled points, the initial/boundary conditions, and the PDE residual. It **does NOT require solution data** obtained from a classic PDE solver (e.g., finite element, finite difference). Our method follows the same setting as PINN.
> > > > > >   - On the contrary, methods like MeshGraphNets and DeepONet **require solution data** from a classic PDE solver as their training data.
> > > > > >   - These two types of methods have different settings and serve different purposes. Therefore, we focus on comparing with PINNs but not methods like MeshGraphNets.
> > > > > >
> > > > > > - We have also updated our previous response to avoid any confusion on “training data”.

---

### Public Comment · ~Yiping_Lu1 · 2022-11-08
**relationship with neural galerkin scheme**

Dear authors

congrats on the nice work. I'm writing this comment aim to understand the relationship between your paper and [1].

[1] Bruna J, Peherstorfer B, Vanden-Eijnden E. Neural Galerkin Scheme with Active Learning for High-Dimensional Evolution Equations[J]. arXiv preprint arXiv:2203.01360, 2022.

---

> ### Author Response · Authors · 2022-11-13
> **Official Comment by Paper943 Authors**
>
> Thanks for bringing this up!
>
> Strictly speaking of technical details, both approaches evolve neural network weights over time. The detailed evolution schemes differ. For example, our formulation facilitates Chorin-style operator splitting while theirs do not.
>
> However, the primary difference is the focus of these two papers.  \[1] targets solving high-dimensional problems that classical solvers often suffer and show that adaptive sampling plays a crucial role in capturing high-dimensional PDEs. By contrast, our work focuses on low-dimensional settings (1D-3D) that heavily rely on classical solvers (e.g., finite element method). We discover that, even on these low-dimensional PDEs, neural-network-based solvers offer a unique advantage of accuracy, memory, and adaptivity at the cost of computing time (under the current standard SGD framework).

---

### Author Response · Authors · 2022-11-13
**General Response by Paper943 Authors**

We thank the reviewers for their insightful responses!

We would like to take this opportunity to emphasize that the primary goal of our work is not extending PINN beyond the training temporal horizon. This is one of the many consequences of our optimization time integrator formulation. Our primary goal is to understand "if we ONLY replace classical solver's spatial representation with a neural network, while keeping the rest unchanged (e.g., time integrator, boundary condition), what tradeoffs do we get?" In this work, we discover that a neural-spatial-representation-based solver offers a unique advantage of accuracy, memory, and adaptivity at the cost of longer computing time (under the current standard neural network training framework).

With that, we are glad that the reviewers find that:

* our manuscript is well-written, easy to follow, and of decent quality (reviewer _7x1Y_, _DwBW_, _uDb6_, _8oNn_, _Tkaf_)
* the proposed idea is interesting, original, and very important for the ML&Physics community(reviewer _7x1Y_, _uDb6_)
* the visualizations of the solution are nicely presented (reviewer _DwBW_)

We address reviewers’ major concerns by:

* significantly improving the related work sections, putting our work into perspective, and adding additional baselines
* adding tables of quantitative results in Appendix D
* including the source code in the supplementary material

Please consider watching the supplementary video in case you haven’t.

We hope the reviewers will consider raising the ratings given these revisions. We look forward to reading any additional feedback.

---

### Decision · Program_Chairs · 2023-01-20

**Decision:**

Reject

**Justification For Why Not Higher Score:**

Limited novelty and insufficient evaluation comparison with alternative methods.

**Justification For Why Not Lower Score:**

N/A

**Metareview: Summary, Strengths And Weaknesses:**

The paper proposes a parametric solver for PDEs that makes use of implicit neural representation (INR)-based models for replacing the usual discrete mesh representations used by classical solvers. The INR parameters are made time dependent.  An initial network is trained using initial and boundary conditions, the parametric approximation of the solution at time t is then input to a solver - as for classical solvers, the differential equation is known - and INR parameters at next time step t+1 are inferred by solving an inverse problem, the procedure is iterated and the INR parameters are thus updated for further solves. The evaluation is performed on a variety of problems.

All the reviewers appreciated the paper but many of them highlighted a limited novelty and asked for more quantitative experiments. The authors did a good job at clarifying some points of the method, they added quantitative evaluations for a better performance comparison with classical solvers and an additional comparison with a related baseline. This was acknowledged by the reviewers but considered as insufficient by several of them for changing their opinion. The paper would benefit from more extensive comparisons with other ML based baselines that also exploit the knowledge of the PDE equation.